# The impact of greenspace or nature-based interventions on cardiovascular health or cancer-related outcomes: A systematic review of experimental studies

Jean C. Bikomeye[1], Joanna S. Balza[1], Jamila L. Kwarteng[2,3], Andreas M. Beyer[3,4], Kirsten M. M. Beyer[1,3]*

1 Division of Epidemiology & Social Sciences, PhD Program in Public and Community Health, Institute for Health & Equity, Medical College of Wisconsin, Milwaukee, WI, United States of America, 2 Division of Community Health, Institute for Health & Equity, Medical College of Wisconsin, Milwaukee, WI, United States of America, 3 MCW Cancer Center, Medical College of Wisconsin, Milwaukee, WI, United States of America, 4 Division of Cardiology, Department of Medicine, Cardiovascular Center, Medical College of Wisconsin, Milwaukee, WI, United States of America

* kbeyer@mcw.edu

**Data Availability Statement:** All data files are available from Figshare: (https://figshare.com/articles/dataset/The_Impact_of_greenspace_or_

## Abstract

### Significance

Globally, cardiovascular disease (CVD) and cancer are leading causes of morbidity and mortality. While having different etiologies, CVD and cancer are linked by multiple shared risk factors, the presence of which exacerbate adverse outcomes for individuals with either disease. For both pathologies, factors such as poverty, lack of physical activity (PA), poor dietary intake, and climate change increase risk of adverse outcomes. Prior research has shown that greenspaces and other nature-based interventions (NBIs) contribute to improved health outcomes and climate change resilience.

### Objective

To summarize evidence on the impact of greenspaces or NBIs on cardiovascular health and/or cancer-related outcomes and identify knowledge gaps to inform future research.

### Methods

Following the Preferred Reporting Items for Systematic Reviews and Meta-Analyses (PRISMA) 2020 and Peer Review of Electronic Search Strategies (PRESS) guidelines, we searched five databases: Web of Science, Scopus, Medline, PsycINFO and GreenFile. Two blinded reviewers used Rayyan AI and a predefined criteria for article inclusion and exclusion. The risk of bias was assessed using a modified version of the Newcastle–Ottawa Scale (NOS). This review is registered with PROSPERO, ID # CRD42021231619.

nature-based_interventions_on_cardiovascular_
health_or_cancer_related_outcomes_A_
systematic_review_of_experimental_studies/
20477121/1); doi: https://doi.org/10.6084/m9.
figshare.20477121.v1.

**Funding:** The work was supported by the following
funding sources: 1. The American Heart
Association (AHA) Scientific focused research
network (SFRN) on disparities in Cardio-oncology
grants: (K.M.M.B: grant ID # 863108; and A.M.B.:
grant ID#: 863107). 2. The AHA SFRN Research
supplement to promote diversity in Science (J.C.B.;
grant ID # 960133) 3. The National Institutes of
Health (NIH) grants: R01HL133029 (A.M.B.); and
R01CA214805 (K.M.M.B) 4. The Medical College of
Wisconsin Cancer Center grant (KM.M.B) 5. The
Medical College of Wisconsin Cardiovascular
center grant 'We Care Fund' (A.M.B.; grant ID #
3308140). The funders had no role in study design,
data collection and analysis, decision to publish, or
preparation of the manuscript.

**Competing interests:** The authors have declared
that no competing interests exist.

## Results & discussion

Of 2565 articles retrieved, 31 articles met the inclusion criteria, and overall had a low risk of
bias. 26 articles studied cardiovascular related outcomes and 5 studied cancer-related out-
comes. Interventions were coded into 4 categories: forest bathing, green exercise, garden-
ing, and nature viewing. Outcomes included blood pressure (BP), cancer-related quality of
life (QoL) and (more infrequently) biomarkers of CVD risk. Descriptions of findings are pre-
sented as well as visual presentations of trends across the findings using RAW graphs.
Overall studies included have a low risk of bias; and alluvial chart trends indicated that NBIs
may have beneficial effects on CVD and cancer-related outcomes.

## Conclusions & implications

*(1) **Clinical implication**:* Healthcare providers should consider the promotion of nature-
based programs to improve health outcomes. *(2) **Policy implication**:* There is a need for
investment in equitable greenspaces to improve health outcomes and build climate resilient
neighborhoods. *(3) **Research or academic implication**:* Research partnerships with com-
munity-based organizations for a comprehensive study of benefits associated with NBIs
should be encouraged to reduce health disparities and ensure intergenerational health
equity. There is a need for investigation of the mechanisms by which NBIs impact CVD and
exploration of the role of CVD biological markers of inflammation among cancer survivors.

## 1. Introduction

Cardiovascular disease (CVD) is the leading cause of global morbidity and mortality [1,2]. In
2019, CVD accounted for approximately 18.6 million deaths globally [3]. In the 2020 Lancet
global burden of disease (GBD) report, ischemic heart disease (IHD) and stroke, both CVD,
were the top-ranked causes of disability adjusted life years (DALYs) in both 50–74 years and
75 years and older age groups [4]; and respectively responsible for 16% and 11% of the total
global deaths, in 2019 [2]. In the US, 126.9 million adults had some form of CVD from 2015
and 2018 [3]. Costs associated with CVD from 2016 to 2017 totaled $363.4 billion ($216.0 bil-
lion in direct costs and $147.4 billion in lost productivity due to morbidity or mortality) [3]. In
addition to the CVD burden, cancer was the sixth leading cause of global mortality in 2019,
and a significant contributor to global morbidity [2]. Further, in 2020 alone, 19.3 million new
cancer cases were diagnosed; and this number is expected to become 28.4 million cases in
2040, a 47% rise from 2020 [5]. There were almost 10.0 million cancer deaths in 2020 [5]. In
2017, the financial burden of cancer in the US was approximately 1.8% of gross domestic prod-
uct or nearly $ 350 billion [6]. The cancer-related healthcare cost was $161.2 billion while the
cost associated with premature mortality was $150.7 billion; and the cost of productivity loss
from morbidity was $30.3 billion [6].

   CVD and cancer have close co-morbid linkages due to multiple shared risk factors [7],
which put cancer survivors at a disproportionate risk for CVD [1,8]. CVD and cancer are
closely linked in a bidirectional causal relationship whereby having one of the diseases puts the
patient at an increased risk of having the other [8,9]. With multiple common risk factors such
as obesity, smoking, and inadequate or low physical activity (PA), co-occurrence of both dis-
eases is a major clinical problem [8]. Each disease affects the treatment of the other, and there-
fore, has a detrimental impact on individual's quality of life (QoL) and survival [8]. For

instance, cancer survivors have increased CVD risk due to cardiotoxic effects of some cancer treatment therapies such as anthracyclines [10,11] and increased risk for CVD mortality [12,13]. Vice versa, there is an increased risk for cancer incidence post CVD diagnosis [14].

Another commonality between CVD and Cancer is how both pathologies are impacted by the environment. In the 2019 GDB risk factor hierarchy, level 1 risk factors include behavioral, environmental or occupational, and metabolic factors [4]. Neighborhood social and built environments, including nature and greenspaces are key determinants of health and important factors in predicting health outcomes [15], including for CVD [16,17] and cancer [18]. Recent estimates suggest that 70% to 80% of CVD burden might be attributable to non-genetic environmental factors, such as lifestyle choices, socioeconomic status (SES), air pollution, lack of neighborhood greenness [19,20] and poorer residential neighborhood characteristics [21]. Neighborhood environmental factors play a key role in influencing obesogenic behaviors [22] such as "food deserts" where grocery stores and food choices are limited [23], and "food swamps" with high concentration of fast-food restaurants selling calorie-dense and nutrient deficient "junk food" with limited healthier food options [24]. Other environmental factors such as limited or poor-quality greenspaces [22,25] and safety concerns [26,27] may reduce use of greenspaces [28,29] and lead to inadequate PA [30]. The double burden of food desertification and food swamps, along with the abovementioned neighborhood-level social risk factors intersect in predisposing individuals to obesity [31]. Inadequate PA and obesity are the two main drivers of high levels of CVD [32,33] and cancer [34] in the US. Additionally, neighborhood disadvantage exposes residents to chronic stress [35] which increases their risk to CVD [36,37] through different biological and pathological processes such as increased levels of cumulative burden of chronic stress and life events, known as allostatic load [38], higher levels of systemic inflammation and differential DNA methylation [39]. On the other hand, neighborhood or community advantage, including increased access to greenspace has been associated with stress reduction [40] as well as weight loss and reduced obesity [41].

Neighborhood material deprivation or neighborhood disadvantage, including reduced neighborhood greenspace quality and quantity, and poor neighborhood social environments have been linked to an increased risk of CVD and cancer [42,43]. For example, in a study with a sample of 25-64-olds in Sweden, among whom 60% had lived at their current addresses for more than five years, neighborhood deprivation, measured by the Care Needs Index [44], was a predictor of CVD risk factors (i.e.: smoking, low PA, and obesity), except for hypertension (HTN) and diabetes that became non-significant in adjusted models [45]. After adjusting for individual level factors (i.e. age, gender, marital status, immigration status, urbanization, and SES), individuals living in highly deprived neighborhoods were significantly more likely to smoke, be physically inactive, and obese, compared to those living in moderately deprived neighborhoods [45]. Similarly, in another sample of Swedish adults aged 25–74 years ($n = 73$ 159), followed from January 1st, 1990, to December 31st, 2008, age-standardized prostate cancer mortality rate was 1.5 times higher in men living in high-deprived neighborhoods than in those living in affluent neighborhoods [46]. Greenspace has been implicated in reducing socio-economic inequities that contribute to neighborhood deprivations [47]. It is important to note that Sweden is more or less of an egalitarian country, which might indicate that these relationships might have higher gradient in countries with high rates of socio-economic inequalities, such as the US [48].

In the US, neighborhood deprivation has been associated with adverse CVD and cancer outcomes [49,50]. In a study with a sample of 25–64 year-olds (1988–1994, $n = 9,961$), residing in a deprived neighborhood increased residents' probability of having an adverse CVD risk profile, independent of individual's SES [49]. Similar findings were observed in both the Jackson Heart Study [43] and the Dallas Heart Study [51]. In the Jackson Heart Study,

neighborhood disadvantages increased CVD risk in a socioeconomically diverse sample of African Americans [43]. For each standard deviation increase in neighborhood disadvantage, CVD risk increased by 25% (hazard ratio = 1.25; 95% confidence interval (CI) = 1.05, 1.49) [43]. In the Dallas heart study, a multilevel regression analysis with a sample of 1174 (18–65 year-olds); found that residing in more deprived neighborhoods was significantly associated with increased BP and incidence of HTN over time during a 9-year period [51]. Individuals living in more deprived neighborhoods had 1.69 times greater odds of developing HTN (OR = 1.69, 95% CI 1.02, 2.02) [51]. Further, in another study, authors used census tract data to investigate the relationship between a 10-year change (1990 to 2000) in neighborhood SES and mortality among 288,555 participating individuals, aged 51–70 years, who enrolled in the National Institutes of Health-AARP Diet and Health Study in 1995–1996 (baseline) and did not move during the study [50]. Mortality data were assessed by linking census tract data to the Social Security Administration Death Master File between 2000 and 2011. Improvement in neighborhood SES was associated with a lower mortality rate, while SES deterioration was associated with a higher mortality rate for both cancer and CVD [50].

Neighborhood built or social environments have been linked with cancer outcomes [18] through multiple studies. In their *"Multi-level Biological and Social Integrative Construct (MBASIC)"* framework, Lynch and Rebbeck integrated macro-environment (i.e.: health care policy, neighborhood, or family structure), individual factors (i.e.: behaviors, carcinogenic exposures, socioeconomic factors, and psychological responses) and biological factors (i.e.: cellular biomarkers and inherited susceptibility variants) to represent the multifactorial and complex nature of cancer etiology [52]. This model has been deemed essential in cancer etiology research [18]. Subsequent research has linked poor neighborhood built and social environments to adverse health outcomes across the entire cancer control continuum including cancer risk [53,54], cancer incidence [55,56], cancer diagnosis [57], cancer treatment [58], cancer survivorship [59], cancer survival [57,60], and cancer mortality [18,61].

In addition to poorly built or social neighborhood environments, global climate change is also adversely impacting health, including poorer CVD and cancer outcomes [62,63]. Extensive literature reviews suggest that increased temperature is associated with higher extreme weather events-related morbidity and mortality, particularly cardiovascular (CV) and respiratory events [64,65]. The higher burden of warmer temperatures on CV health includes increased risk of myocardial infarction (MI) [66] and mortality for IHD in North America [67]. A 2008 study found that for every increase of 4.7˚C in mean daily temperature, there was a 2.6% increase in CV mortality in California (95% confidence interval (CI): 1.3, 3.9) [67].

Greenspace is a major component of the built neighborhood environment and has been linked with increased neighborhood property values [68–70]. Additionally, greenspace has been linked with many positive health outcomes [71], including lower odds of being overweight or obese, a major risk factor for both CVD and cancer [41]. Some of empirically investigated benefits of greenspace on CV health include increased angiogenic capacity [72], reduced CVD risk [17,73], decreased CVD morbidity [74], and decreased CVD mortality [19,75,76]. Similarly, some of the benefits of greenspace on cancer outcomes include enhanced cancer prevention initiatives [77,78], reduced cancer incidence [78,79], improved cancer survivorship [78,80], and reduced prostate cancer mortality [81]. Additionally, greenspace helps in sequestering carbon and contributing to greenhouse gases reduction, therefore is a viable intervention for the adverse impacts of climate change on both environmental and human health [82].

There is growing literature evidence on the impact of greenspace on improving clinical outcomes in CVD and cancer patients through different interventions such as "park prescription" programs and other nature-based interventions (NBI) [83–86]. Some of this evidence was found through experimental studies, suggesting possible causal relationships, and

opportunities for specific interventions to improve CVD and cancer-related health outcomes. However these experimental studies have not yet been systematically reviewed to bring all existing evidence together [1]. In this review, we sought to systematically summarize findings from experimental studies with greenspace interventions and identify potential literature gaps for future research. We use an expanded definition of greenspace exposure that include forest bathing, nature viewing, nature visit, parks visits, gardening, etc. We conducted a systematic review of studies that have investigated the impact of greenspace or NBI on two main health outcomes: CVD and cancer. CVD outcomes include morbidity and mortality across different CVD conditions. Cancer-related outcomes include different measures across the cancer control continuum including cancer risk, prevention, detection, diagnosis, treatment, survivorship, end of life or mortality, as well as cancer-related QoL.

## 2. Methods

This review followed a pre-defined protocol that was developed following the preferred reporting items for systematic review and meta-analysis protocols (PRISMA-P) statement and checklist [87,88]; and was pre-registered with PROSPERO, ID # CRD42021231619. This review then followed the PRISMA 2020 reporting guidelines [89]. The PRISMA chart is illustrated in Fig 1; and the PRISMA 2020 27-items checklist is annexed in *Appendix A*.

### 2.1. Literature search

A comprehensive literature search was developed in collaboration with a medical librarian and peer reviewed using the Peer Review of Electronic Search Strategies (PRESS) guideline [105]. The following citation databases were searched on March 10[th], 2021: Web of Science, Scopus, Medline, APA PsycINFO, and GreenFile. Searches were limited to articles written in English. Databases were chosen because we sought to include all citation databases of peer-reviewed literature with comprehensive citation data for many different academic disciplines (Web of Science), source neutral literature curated by independent subject matter experts (Scopus), medical sciences from the National Library of Medicine's bibliographic database (Medline), literature in the field of psychology (PsycINFO) and literature focused on nature or greenspace

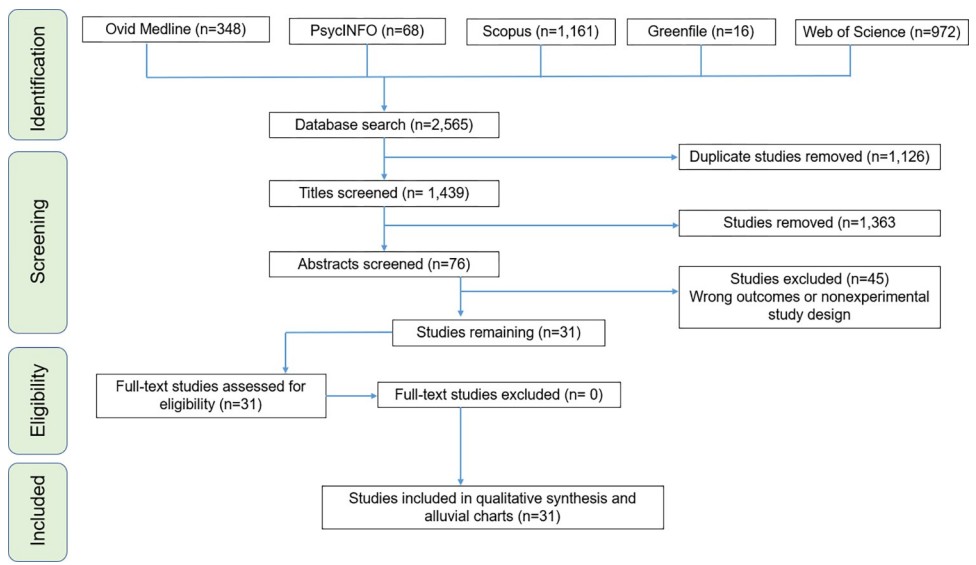

**Fig 1. Graphical illustration of PRISMA 2020 guidelines in articles' selection process.**

(GreenFile). Search strategies were created using medical subject headings (MeSH) and keywords combined with database-specific advanced search techniques. MeSH terms and keywords were identified to represent greenspace interventions, CVD, and cancer. Keywords related to greenspace or NBI (i.e.: park prescription, wilderness therapy, forest bathing, forest therapy, green exercise, etc.), CVD outcomes (i.e.: heart failure (HF), stroke, coronary artery disease (CAD), MI, cardiac arrest, major adverse CV event (MACE), etc.) and cancer-related outcomes (i.e.: cancer prognosis, cancer incidence, cancer mortality, etc.). A full search strategy is annexed in *Appendix B.*

A total of 2565 results from literature searches (Medline: 348, PsycINFO: 68, Scopus: 1161, Web of Science: 972 and Greenfile: 16) were downloaded into EndNote where duplicate articles (*n* = 1126) were removed. 1439 unique publications were uploaded into Rayyan AI, an online tool for systematic review [90,91] available at https://www.rayyan.ai/. The web app facilitated article screening and eased collaboration between two independent reviewers.

### 2.2. Article selection process

The following PICO framework [92] of inclusion and exclusion criteria was followed:

*P (Population)*: No restriction. All ages, genders, races/ethnicities, healthy or diseased individuals are included.

*I (Interventions)*: *Exposure to* greenspace or NBIs such as forest bathing, greening exercise, nature viewing, or gardening.

*C (Comparison)*: All types of controls, or simple pre-post experiments without formal controls

*O (Outcomes)*: CVD or cancer-related outcomes

1. CVD related outcomes include BP and MACE, as defined in previous studies [93–95] including: occurrence of fatal and nonfatal MI, HF, cerebrovascular disease or CV accident or stroke (fatal and nonfatal), or coronary artery bypass grafting (CABG) and cardiac arrest. Both preventive measures (indictors of good CV health among healthy individuals) and restorative measures (indicators of improved CV health among individuals with CVD) are all considered.

2. Cancer-related outcomes include lifestyle changes (i.e., gardening continuation after intervention) and QoL during cancer survivorship, and cancer outcomes (i.e.: cancer prognosis, cancer incidence, cancer mortality, etc.). We used the National Cancer Institute definition of cancer survivorship in defining the cancer survivor's population which proposes that survivorship starts the first time the patient was told by a healthcare provider that they have cancer until the end of life [96].

Since the overall goal of the review is to look at the impact of interventions on outcomes, Using Rayyan, search results were systematically screened by two reviewers (J.C.B and J.S. B) to determine eligibility. Reviewers first screened articles' titles against eligibility criteria, excluding any article that did not clearly meet the PICO criteria by reading articles' titles. Conflicts were resolved and the process was repeated, screening full abstracts, and then article's methods section. If a conflict could not be resolved between the two reviewers, a third mediator (KMMB) was consulted. Finally, the reference lists of all included articles were screened to identify relevant publications not retrieved by electronic database searches.

### 2.3. Eligibility criteria

Inclusion and exclusion criteria are summarized in Table 1.

**Table 1. Inclusion and exclusion criteria.**

| Article inclusion criteria | Article exclusion criteria |
|---|---|
| Experimental (with or without control and quasi-experimental) studies. | Reviews, protocols, case reports, and commentaries were excluded. |
| Studies with human participants. | Studies without human subjects (animal studies) |
| Articles published in English | Articles published in any language other than English |
| Studies that look at CV health or cancer-related outcomes | Studies looking at any outcome other than those related to CV health or cancer |
| Exposure to greenspace or NBI (predictor variable) | Studies that did not have greenspace or NBI as a predictor variable. |
| Studies with pre/post, quasi-experimental or experimental design | Any study types other than pre/post, quasi-experimental or experimental |
| Studies with available full text | Studies without full text availability |

## 2.4. Data extraction and reporting

Extracted data are summarized in Tables 3 and 4 and include: (1) Studies' geographical information (City, state, country); (2) Studies' urbanicity setting (rural, semi-urban, or urban) where applicable; (3) type of greenspace or nature-based interventions + controls description where applicable, (4) assumptions made or hypotheses; (5) Measures of any CVD related outcome (incidence, morbidity, or CVD related mortality); (6) Measures of any cancer-related outcome (anything from the cancer control continuum, cancer-related quality of life (QOL), or cancer-related mortality; (7) cancer type under investigation (specific or any type); (8) Covariates adjusted for including (a) individuals level variables such as demographic information (when available); socioeconomic information (when available); comorbidity information (when available); and (b) neighborhood factors (when available) such as social environment factors, and other neighborhood-built environment characteristics; (9) Statistical analyses conducted; (10) Studies strengths and weaknesses". We used the following information to create alluvial charts as a visual representation of trends across studies by outcomes of interest, a method that was previously used in previous systematic reviews [97]:

1. Article reference

2. Study country

3. Intervention type

4. CVD outcomes or cancer-related outcomes

5. Conclusion (weather a statistical test found the intervention to be significantly beneficial: *Beneficial effect*, or no statistically significant difference between control and experimental groups: *Not significant*; or whether beneficial changes were observed in the control group instead of in the interventional or experimental group: *Significant in controls*.

Two excel datasets used to create alluvial charts for (1) CVD, and (2) cancer-related outcomes are respectively annexed in *Appendices C1 and C2*.

## 3. Results

### 3.1. PRISMA 2020 chart illustrating our articles screening process

From 2,565 articles initially retrieved from database searches, 31 articles meeting our pre-defined criteria remained after screening, as illustrated in Fig 1. At the abstract screening stage, 45 articles were excluded because they did not meet at least one of our pre-defined

**Table 2. Risk of bias assessment of included studies using a Modified version of the Newcastle–Ottawa Scale.**

| | Study: Author (year) | Final score | SELECTION | | | | COMPATIBILITY | OUTCOMES | | |
|---|---|---|---|---|---|---|---|---|---|---|
| | | | Representativeness of exposed group | Non-exposed group | Ascertainment of exposure | Baseline difference | Groups compatibility | Outcome assessment | Exposure duration | Groups follow up |
| 1 | Mao et al., 2012 [100] | 9 | 1 | 1 | 1 | 1 | 2 | 1 | 1 | 1 |
| 2 | Navalta et al., 2019 [101] | 6 | 1 | 1/2 | 1/2 | 0 | 1 | 1 | 1 | 1 |
| 3 | Engell et al., 2020 [102] | 5 | 0 | 1/2 | 1/2 | 0 | 1 | 1 | 1 | 1 |
| 4 | Duncan et al., 2014 [103] | 6.5 | 1 | 1/2 | 1/2 | 1/2 | 1 | 1 | 1 | 1 |
| 5 | Furuyashiki et al.,2019 [104] | 4 | 1 | 0 | 0 | 0 | 0 | 1 | 1 | 1 |
| 6 | Grazuleviciene et al., 2016 [105] | 9 | 1 | 1 | 1 | 1 | 2 | 1 | 1 | 1 |
| 7 | Lee et al., 2011 [106] | 6 | 1 | 1/2 | 1/2 | 0 | 1 | 1 | 1 | 1 |
| 8 | Li, et al., 2016 [107] | 5 | 0 | 1/2 | 1/2 | 0 | 1 | 1 | 1 | 1 |
| 9 | Mao et al., 2017 [108] | 9 | 1 | 1 | 1 | 1 | 2 | 1 | 1 | 1 |
| 10 | Mao et al., 2012 [109] | 9 | 1 | 1 | 1 | 1 | 2 | 1 | 1 | 1 |
| 11 | Niedermeier et al., 2017 [110] | 6 | 1 | 1/2 | 1/2 | 0 | 1 | 1 | 1 | 1 |
| 12 | Chen et al., 2018 [111] | 4 | 1 | 0 | 0 | 0 | 0 | 1 | 1 | 1 |
| 13 | Ochiai et al., 2015 [112] | 4 | 1 | 0 | 0 | 0 | 0 | 1 | 1 | 1 |
| 14 | Peterfalvi et al., 2021 [113] | 4 | 1 | 0 | 0 | 0 | 0 | 1 | 1 | 1 |
| 15 | Pretty et al., 2005 [114] | 8 | 1 | 1 | 1 | 1 | 1 | 1 | 1 | 1 |
| 16 | Song et al., 2018 [115] | 6 | 1 | 1/2 | 1/2 | 0 | 1 | 1 | 1 | 1 |
| 17 | Tsutsumi et al., 2017 [116] | 4 | 1 | 0 | 0 | 0 | 0 | 1 | 1 | 1 |
| 18 | White et al., 2015 [117] | 6 | 1 | 1/2 | 1/2 | 0 | 1 | 1 | 1 | 1 |
| 19 | Bielinis et al., 2019 [118] | 4 | 1 | 0 | 0 | 0 | 0 | 1 | 1 | 1 |
| 20 | Yu et al., 2017 [119] | 4 | 1 | 0 | 0 | 0 | 0 | 1 | 1 | 1 |
| 21 | Koura et al., 2016 [120] | 4 | 1 | 0 | 0 | 0 | 0 | 1 | 1 | 1 |
| 22 | McEwan et al., 2021 [121] | 6 | 1 | ½ | 1/2 | 0 | 1 | 1 | 1 | 1 |
| 23 | Park et al., 2017 [122] | 7.5 | 1 | 1 | 1/2 | 1 | 1 | 1 | 1 | 1 |
| 24 | Song et al., 2013 [123] | 6 | 1 | 1/2 | 1/2 | 0 | 1 | 1 | 1 | 1 |
| 25 | Wu et al., 2020 [124] | 9 | 1 | 1 | 1 | 1 | 2 | 1 | 1 | 1 |

*(Continued)*

**Table 2.** (Continued)

| | Study: Author (year) | Final score | SELECTION | | | | COMPATIBILITY | OUTCOMES | | |
|---|---|---|---|---|---|---|---|---|---|---|
| | | | Representativeness of exposed group | Non-exposed group | Ascertainment of exposure | Baseline difference | Groups compatibility | Outcome assessment | Exposure duration | Groups follow up |
| 26 | Lanki et al., 2017 [125] | 6 | 1 | 1/2 | 1/2 | 0 | 1 | 1 | 1 | 1 |
| 27 | Bail et al., 2018 [126] | 7 | 1 | 1 | 1 | 0 | 1 | 1 | 1 | 1 |
| 28 | Blair et al., 2013 [127] | 4 | 1 | 0 | 0 | 0 | 0 | 1 | 1 | 1 |
| 29 | Demark-Wahnefried et al., 2018 [128] | 8 | 1 | 1 | 1 | 1 | 1 | 1 | 1 | 1 |
| 30 | Li et al., 2008 [129] | 6 | 1 | ½ | ½ | 0 | 1 | 1 | 1 | 1 |
| 31 | Li et al., 2007 [130] | 4 | 1 | 0 | 0 | 0 | 0 | 1 | 1 | 1 |
| | Average (±SD) Score | 6.0 (±1.8) | | | | | | | | |

Item assessment description:

Representativeness of exposed group: One star was given if the study population reflected the title or abstract of the article (i.e., the group is representative (or somewhat representative of the community average). For example, a study that only used male subjects, but the title/abstract did not specify that the 'community' was males (leaving room for confusion), did not receive a star. However, a study that said in the title they were assessing results in a "population of healthy young males" and then used healthy young males, did receive a star.

Non-exposed group: One star was given if two groups (exposed or not exposed) were drawn from the same population. Half of a star was given if the same group served as the control group (on a different day or time in which they were not exposed to the intervention). No star was given if it was a simple pre-exposure and post-exposure measurement with no control group OR if the two groups (exposed or not exposed) were not drawn from the same population.

Ascertainment of exposure: One star was given for studies where participants were randomly assigned to be in control or exposure group. Half of a star was assigned if the same sample was the control one day then the experiment another day or time, or if the two groups were similar but not random. No star if there was no control group.

Baseline difference: One star was given if there was a control group, and there was no baseline difference. Half a star was given if the same group served as their own controls, by repeating the experiment twice, once with exposure and once without (as a control) and there was no difference at baseline. No star given if there was no control group, there were differences between the group at baseline, or if this was not reported.

Compatibility and controlling factors between groups: If the study design controlled for two or more factors in both groups that may have impacted the outcome (i.e., diet, caffeine, sleep) they were given 2 stars. If they had two compatible groups but controlled for only one or no factors, they were given one star. No star was given for simple preexposure/postexposure tests with no control group.

Outcome assessment: One star was given if the study clearly defines outcomes and how they were assessed.

Exposure duration: One star was given if raters perceive that exposure duration was long enough to observe differences in outcomes.

Cohorts follow up: One star was given if all subjects were followed up until completion or if there if raters perceive the number of subjects lost to follow as small enough to not introduce any bias

inclusion criteria. Each one of the excluded studies was either not experimental, or not looking at one of the outcomes of interest.

## 3.2. Risk of bias assessment

The risk of bias for 31 included studies was assessed using a modified version of Newcastle–Ottawa Scale (NOS). Two reviewers (J.C.B. and J.S.B.) independently assessed articles on eight pre-defined items including representativeness of exposed cohort, similarity of cohorts' origins, similarity of exposed vs non-exposed cohorts (compatibility), ascertainment of exposure, baseline differences, outcome assessment, exposure duration (enough to observe outcome),

**Table 3. Characteristics for 26 studies with Cardiovascular outcome.**

| Author and year | Country, City, State Urbanicity setting | Sample Size | Study type | Follow up duration | Age (Mean ± SD, years) | Intervention Green space exposure | Exposure description + Greenspace type and Control group | Hypothesis/ Assumption | Covariates | CV related outcome | Statistical Analyses | Findings | Strengths & Weaknesses Conclusions |
|---|---|---|---|---|---|---|---|---|---|---|---|---|---|
| Mao et al., 2012 [100] | Hangzhou city, Zhejiang Province, China Urban | (n = 24): 12 for both the control and the experimental groups | Experimental study | 7-day duration from 23 to 30 July 2011 | Age 60 to 75 years | Forest bathing Two daily pre-determined unhurried pace walks for 1.5h with 20 minutes rest during the walk, one in the morning and another one in the afternoon Participants had a pre-determined daily schedule for 7 days | A broad-leaved evergreen forest experience whereby predominant species are Ormosia hosiei, Cinnamomum camphora, Magnolia officinalis subsp. officinalis subsp. biloba, and Nyssa sinensis. The control group was sent to an urban area | There is a therapeutic effect of forest bathing on hypertension in elderly subjects. | Demographic: Age, body mass index Socioeconomic: N/A Comorbidity: N/A Environmental factors: N/A | HTN BP indicators, CV disease-related pathological factors including endothelin-1, homocysteine renin, angiotensinogen, angiotensin II, angiotensin II type 1 receptor, angiotensin II type 2 receptor as well as inflammatory cytokines interleukin-6 and tumor necrosis factor alpha (TNF-α) | Kolmogorov-Smirnov test and Levene's test were respectively used for normality and homogeneity of variances. t-test for comparison between two groups Mann-Whitney U test or Wilcoxon Signed Ranks test for two independent or related samples | No baseline difference in all biomarkers investigated. Participants who experienced a 7-day forest bathing trip showed a significant decrease in systolic BP (SBP) and diastolic BP (DBP) compared with that of the city group Pulse pressure decreased No change in heart rate (HR) | Limitation in participants size and age range Forest bathing has therapeutic effects on HTN reduces BP and prevents CV disorders |
| Navalta et al., 2019 [101] | USA State and city not specified Urban | 10 (7 males and 3 females) | Experimental study | 30 mins | Age 29.2 (± 7.3) | Walk in green and brown environments | 30-min self-paced walking (WALK) in: indoor, outdoor urban, green, and two brown environments No control group (use of repeated measures in different environments) | Exercise in a natural setting would provide similar beneficial physiological and perceptual effects. | Demographic factors: Age, height, and mass Socioeconomic factors: N/A Comorbidity: N/A Environmental factors: N/A | HR, SBP, and measures of stress, comfort, and calm | Analysis was done with a 3 (Time: Pre-Sit, Post-Sit, Post-Walk) X 5 (Environment: indoor, urban, green, brown, brown below sea level) analysis of variance (ANOVA) with repeated measures on both factors. | HR was elevated in urban vs green (p = 0.05) SBP was lower after SIT compared to PRE and WALK (p = 0.05) | Limitation lie on the focus only on the student population and small number of population, budget limitations. The study was experimented in a natural setting which includes ambient noise, the presence of non-study personnel, past memories of visits to the particular setting, physical discomfort, and odors Exercise in a desert environment is as beneficial as exercise in a green environment. |
| Engell et al., 2020 [102] | Norway Urbanicity not specified | 9 male students | Experimental study | 7 days | Age: mean (SD) = 23.55 (± 2.34) | View of a modest natural environment while resting after physical exertion | A window view with a forest dominated hillside and field land. The field land and forest were fully or partially covered in snow in all sessions. No control group (All participants were engaged in the same activity) | Three hypotheses: 1) Resting with a window view of a natural environment improves cognitive function 2) Taking a break in front of a window, viewing a natural environment after minor physical activity produces more efficient heart rate restoration. 3) Taking a break in front of a window, seeing a natural environment after minor physical activity causes reduced heart rate responses. | demographics and potential confounders (amount of sleep, mood state, current health, exercise history current week, consumption of potentially confounding substances) Socioeconomic factors: N/A Comorbidity: N/A Environmental factors: N/A | Measures of choice reaction time (CRT) and HR variability (HRV): intervals between successive heartbeats. | Within-subjects repeated measures Wilcoxon signed-ranks test with rank-biserial correlation | Improvement in CRTs and HR restoration after resting with a window view, compared to resting without a view Effect of greater effect of cognitive enhancement and physiological restoration in resting after exercise with view to natural environment. | Limitation: Modest sample size Cognitive enhancement and physiological restoration after exercise in resting with a view of natural environment compared to resting without this view. |

*(Continued)*

**Table 3.** (Continued)

| Author and year | Country, City, State Urbanicity setting | Sample Size | Study type | Follow up duration | Age (Mean ± SD, years) | Intervention Green space exposure | Exposure description + Greenspace type and Control group | Hypothesis/ Assumption | Covariates | CV related outcome | Statistical Analyses | Findings | Strengths & Weaknesses Conclusions |
|---|---|---|---|---|---|---|---|---|---|---|---|---|---|
| Duncan et al., 2014 [103] | Coventry, UK Urban | 14 children (7 boys, 7 girls | Experimental study | 15 min | 10 (± 1) | Exercise in green environment Green exercise | Control condition: viewing a blank screen. Experiment: watching a film of cycling in a forest environment (Through the Forest; World Nature Video, Lunteren, The Netherlands) Participants in control scenario cycled while viewing a blank screen under moderate intensity of 15 min. | Changes in BP, HR, and mood state responses are due to the results of viewing a video depicting green exercise | Demographic factors: Gender, age Socioeconomic factors: N/A Comorbidity: N/A Environmental factors: N/A | BP, HR, and mood state responses Pre-, immediately post-exercise and 15 min post-exercise | Paired samples t-tests: used to study baseline differences between groups ANOVA with Bonferroni post-hoc pairwise comparisons Effect size was measured with partial eta squared (η2) | Lowered SBP in green exercise compared to control condition No difference in DBP Higher HR in all conditions Mood state of Fatigue is higher while vigor is lower | Limitation was that the study was exploratory which lead to weak statistical power, the difficulty of identifying the hypotensive effects. Hypotensive effect for children following green exercise compared to exercise alone. |
| Furuyashiki et al., 2019 [104] | Hiroshima City, Japan Urban | 155 | Experimental study | 16 –one day long sessions for 3 years (2012–2014) | Age range: 19–59 Mean 44.0 (± 9.6) | Forest bathing | Within a national park. Vegetation: natural forests with a temperate climate No control group. Authors did measurements of outcomes indicators before and after a forest bathing intervention | There are physiological and psychological effects of forest bathing on people of working age with and without depression tendencies. | Demographic factors: Age, Sex, Body mass index, Medication, Health-related QOL. Socioeconomic factors: N/A Comorbidity: N/A Environmental factors: N/A | The circulatory functions of SBP, DBP, and pulse rate (PR) | Shapiro–Wilk test for confirmation of data normality. t tests, Wilcoxon signed rank test, Chi-square test, Mann–Whitney U test, simple regression analysis, and Spearman's rank correlation coefficient | Reduction in SBP, DBP, and in negative profile of mood states (POMS) items after a forest bathing session Before forest bathing, those with depression expressed POMS negative items than those without depressive tendencies. After forest bathing, there is improvement in many POMS items | The limitations are: Statistical significance is found only on those with depression; there was a short time of experiment in a single day and the paucity of research to cross section research to compliment the effects. Forest bathing has a positive effect on mental health, especially among those with depression |
| Grazuleviciene et al., 2016 [105] | Kaunas City, Lithuania Urban | 20 male and female half in experimental and half in control groups | Experimental study | 7 days | 45–75 years Mean: 62.3 (± 12.6 years) | Green exercise: City Park or urban street environment | Greenspace exposure: urban park environment (pine park terrain) Control group was exposed to an urban street environment | Walking in a park has a more beneficial effect on CAD patients' stress measures and heart function than walking in a city. | Demographic factors: Gender, age, BMI Socioeconomic factors: N/A Comorbidity: N/A Environmental factors: N/A | Hemodynamic parameters: HR, SBP, and DBP physiological measure of stress: Cortisol levels mood scores Feelings and emotional state | Normal distribution of variables and its logarithmic transformations were tested by using the Shapiro-Wilk test Unpaired and paired t-tests were used to compare the means. Wilcoxon signed-rank test (within-subjects comparisons) and exact Mann-Whitney U test (between subjects' comparisons) | Greater Reduction of cortisol levels (stress) in city parks than urban streets Reduction in DBP in the park | Limitations are small sample size, small treatment and non-identified mechanisms through greenspace reduce stress and enhance cardiac functions PA in greener environment with less noise and polluted air has positive effect on CAD patients stress level and hemodynamic parameters |
| Lee et al. 2011 [106] | Hokkaido, Japan Urban | 12 males half in experimental and half in control groups | Experimental study | 15-min exposure to forest or urban environmental stimuli (observation period) 3 day– 2-night field experiment | 21.2 (± 0.9) years | Forest bathing and urban control: 15 minutes of viewing an urban or forest stimuli 12–14th September 2006 hotel stay whereby potential confounders were controlled for (food, drinks, and PA) | Forest: broad-leaved deciduous trees Control group was assigned to an urban environment (commercial area) | Natural habitats, such as forests, have a substantial favorable association with human health. | Demographic information: Age, BMI Socioeconomic factors: N/A Comorbidity; Past and current mental disorders, cardiovascular and allergic diseases Environmental factors: N/A | HR Salivary cortisol level, PR and feelings | Paired t tests to compare groups' differences Wilcoxon signed rank test for verification of statistical differences in psychological indices. | Increased parasympathetic nervous activity and suppresses sympathetic activity of forest bathing participants compared with the urban environment Reduced Salivary cortisol level and PR in forest than urban Forest bathing enhances positive feelings | Limitations: small sample size, the focus on male gender, which lead to the failure of generalizing results to women. There are positive effects of forest bathing on physical and mental health, thus health promotion |

(Continued)

**Table 3.** (Continued)

| Author and year | Country, City, State Urbanicity setting | Sample Size | Study type | Follow up duration | Age (Mean ± SD, years) | Intervention Green space exposure | Exposure description + Greenspace type and Control group | Hypothesis/ Assumption | Covariates | CV related outcome | Statistical Analyses | Findings | Strengths & Weaknesses Conclusions |
|---|---|---|---|---|---|---|---|---|---|---|---|---|---|
| Li et al., 2016 [107] | Agematso, Nagano Prefecture, Japan Urban | 19 males | Experimental study | 4 weeks | 51.2 (± 8.8) | Forest bathing Walking for 2.6 km for 80 min each in both morning and afternoon on Saturdays | Forest Environment: forest park The control group was sent to the urban region with no trees | Walking in a forest environment would improve cardiovascular function. | Demographic factors: Age, Height (cm), Body weight (kg) BMI Socioeconomic factors: Smoking status Comorbidity: N/A Environmental factors: N/A | CV parameters: BP and PR Mood states (POMS) Metabolic parameters: Urinary adrenaline; Urinary dopamine Serum adiponectin | Paired t test | Forest bathing reduces PR, increases vigor, and decreases depression, fatigue, anxiety, and confusion After forest bathing, there is decrease of Urinary adrenaline and Urinary dopamine compared to urban walking The increase in Serum adiponectin | The limitation is that the order of exposure was not corrected or counterbalanced; sample size is not representative. Forest bathing has a positive effect on health and physiological and psychological relaxation |
| Mao et al., 2017 [108] | Hanhzhou City, China Urban | 33 Forest Group (n = 23) City Group (n = 10) | Experimental study | 4 days: From 20 to 24 August 2015 | Forest Group 72.86 (± 5.85) City Group 70.70 (± 3.68) | Forest bathing | Forest: predominant species are pine, China fir, and bamboo Confounders such as (food and drinks intake, smoking and PA were controlled for. The control group was sent to urban region or city region in downtown area of Hangzhou | Forest bathing is thought to be beneficial to CAD patients, such as those with chronic heart failure, and may even provide therapeutic advantages. | Demographic factors: Age, Gender, Hight, Weight, BMI Socioeconomic factors: New York Heart Association Class Comorbidity: N/A Environmental factors: N/A | Chronic HF Biomarkers for HF BNP and NT-Pro BNP, CV disease- related factors Oxidative indicators Profile of Mood States Air quality | Kolmogorov-Smirnov test and Levene's test were respectively used for normality and homogeneity of variances. t-test for comparison between two groups Mann–Whitney U test or Wilcoxon Signed Ranks test for two independent or related samples Chi-squared test for count data Kruskal-Wallis test for multi-group comparisons with post hoc Bonferroni adjustment | Forest bathing decreases brain natriuretic peptide (BNP), and components of the renin-angiotensin system (RAS) including renin, angiotensinogen (AGT), angiotensin II (ANGII), and ANGII receptor type 1 or 2 (AT1 or AT2), inflammatory cytokines including nterleukin-6 (IL-6) and TNF- α, and other markers of oxidative stress | Limitations are small sample size, indicators were measured in a specific time, climatic factors were not considered Forest bathing has a therapeutic role for CV disorders |
| Mao et al., 2012 (2) [109] | Zhejiang, China Urban | 20 male university students | Experimental study | 2 day | 20.79 (±0.54 years) | Forest bathing | Broad-leaved evergreen forest with urban area controls The control group was sent in urban city | There are yet to be any direct demonstration of whether forest bathing has any other health benefits. | Demographic factors: Age, weight, BMI Socioeconomic factors: N/A Comorbidity: N/A Environmental factors: N/A | Serum total SOD Lipid peroxidation (malondialdehyde) Serum and plasma Serum cortisol and testosterone Lymphocyte assay The profile of mood states (POMS) | Kolmogorov-Smirnov test and Levene's test were respectively used for normality and homogeneity of variances. t-test for comparison between two groups Mann–Whitney U test for non-normally distributed data | There were no differences in baseline values for all biomarkers between the two groups There's reduction of oxidative stress and pro-inflammatory level to those exposed in forest Serum cortisol levels were lower for the forest group than those of urban Concentration of plasma endothelin-1 (ET-1) was lower than those in forest group Increased vigor after exposure to forest and POMS lower after the forest exposure | Limitations: small sample size, results don't reflect in old or infirm people, climatic data such as air pollution, air quality not considered Forest bathing has benefits to human health |

*(Continued)*

**Table 3.** (Continued)

| Author and year | Country, City, State Urbanicity setting | Sample Size | Study type | Follow up duration | Age (Mean ± SD, years) | Intervention Green space exposure | Exposure description type and Control group | Hypothesis/ Assumption | Covariates | CV related outcome | Statistical Analyses | Findings | Strengths & Weaknesses Conclusions |
|---|---|---|---|---|---|---|---|---|---|---|---|---|---|
| Niedermeier et al., 2017 [110] | Innsbruck, Austria Urban | 42 | Randomized trial study | 3 hours | Age 32.0 (± 12.0) | Green exercise Mountain hiking | Three-hour green exercise intervention (mountain hiking) The control group: Sedentary control condition was in quiet room | Mountain hiking has the effects of longer-duration physical exercise sessions. | Demographic factors: Age, weight, BMI, Physical activity, Mountain tours Socioeconomic factors: N/A Comorbidity: N/A Environmental factors: N/A | Endocrine and CV physiological measures: Salivary cortisol concentration, HRV and BP. | Repeated measures ANOVA | No differences were found between mountain hiking and treadmill walking in salivary cortisol Salivary cortisol decreased in all conditions, but showed a larger decrease after both mountain hiking and treadmill walking compared to the sedentary control situation Salivary cortisol changes from baseline to follow-up did not significantly differ between the three conditions for HRV and BP | Limitations: low statistical power, focus on male gender, focus on Japanese, cross/ section design/design without control intervention, intensity duration impacts the results Hiking indoors or outdoors has effects on salivary cortisol concentration There are Environmental effects on salivary cortisol, BP, and HRV. |
| Chen et al., 2018 [111] | Nantou, Taiwan Urban | 16 female | Pre-test and posttest experimental design | 2 day | 46.88 (± 7.83 years) | Forest bathing Two-day (one-night) forest therapy program | Natural scenery, such as broad-leaved trees and waterfalls No control groups. The study used a preset and posttest design | Visiting a forest, in addition to receiving medical care from doctors, may help middle aged women improve their psychological and physiological wellbeing. | Demographic factors: Age. Socioeconomic factors: N/A Comorbidity: N/A Environmental factor: N/A | Psychological factors Profile of Mood States (POMS) State anxiety and trait anxiety Physiological measurement: PR, SBP, and DBP | Descriptive analysis, and a series of paired samplet tests | Negative mood states (i.e., confusion, fatigue, anger-hostility, and tension) and anxiety levels decreased after forest visit Vigor improved after the program Decrease in systolic BP after the program | Limitations: Environmental factors not considered, menopause factor limited the study findings, substance uses not controlled, not control group was in the study. Forest bathing has good effect mental health and systolic BP among the middle-aged female group |
| Ochiai et al., 2015 [112] | Agematsu, Nagano Prefecture, Japan Urbanicity not specified | 9 male | Experimental study | 1 day | 56 (± 13.0) | Forest bathing | Natural forest No control groups. Participants were sent in the forest for therapy and pre-treatment measures were taken | Forest therapy may have physiological and psychological impacts on middle-aged men with high-normal blood pressure. | Demographic factors: Age Socioeconomic factors: N/A Comorbidity: N/A Environmental factors: N/A | SBP, DBP Urine and blood samples Semantic Differential (SD) method Profile of Mood State (POMS) | Paired sample t-tests were used to compare physiological indices Wilcoxon signed-rank test were used to compare psychological test results before and after forest bathing | SBP, DBP, urinary adrenaline, and serum cortisol were lower after forest therapy Relaxing and feeling natural tension-anxiety, confusion, and anger-hostility lower after forest therapy | Limitation: lack of control group, results not extrapolated to female or hypertensive adults Forest bathing reduces BP and prevents clinical HTN |
| Peterfalvi et al., 2021 [113] | Pécs, Hungary Urban | 12 | Pretest-posttest field experiment | 2 day | 38.5 | Forest bathing | 2-h leisurely forest walking in recreational woodland area with oak forest with sub-Mediterranean features and additional tree species Diverse vegetation including several species of shrubs and herbaceous plants fading wild garlic No control group. The study used a pretest and posttest field experiment in which participants were sent in the forest. | A single session of 2-hour forest bathing in the adjacent forests affects the quantity and function of CD8 + T cells, NK and NKT cells, as well as the cardiovascular effects in working-age persons. | Demographic factors: Gender, age, BMI, Socioeconomic factors: smoking, working status Comorbidity: N/A Environmental factors: N/A | SBP | Data normality test Paired samplest t-test and Wilcoxon test were used for pre-post statistical significance parameters Independent samples t-test and Mann–Whitney test were used for the comparison of the seasons' pre (basal level) parameters. | Decrease of SBP after the trips both in late spring and in the winter | Limitation: no forest air samples were collected, small sample size of participants, no observation of duration of forest walking effects, no identical experiment in non-forested city area, and no analysis of forest air composition Forest has medicinal potential |

*(Continued)*

Table 3. (Continued)

| Author and year | Country, City, State Urbanicity setting | Sample Size | Study type | Follow up duration | Age (Mean ± SD, years) | Intervention Green space exposure | Exposure description + Greenspace type and Control group | Hypothesis/ Assumption | Covariates | CV related outcome | Statistical Analyses | Findings | Strengths & Weaknesses Conclusions |
|---|---|---|---|---|---|---|---|---|---|---|---|---|---|
| Pretty et al., 2005 [114] | Colchester, UK Urbanicity not specified | 100 55 female, 45 males | Experimental study | | 24.6 (±0.99) | Green exercise running without exposure to scenery images | Randomized exposure to a sequence of 30 scenes projected on a wall whilst exercising. The scenes were categorized as rural pleasant, rural unpleasant, urban pleasant and urban unpleasant The control group was set to run without being exposed to any visuals. | There may be a synergistic benefit to engaging in physical activities while exposed to nature. | Demographic factors: Age Socioeconomic factors: N/A Comorbidity: N/A Environmental factors: N/A | BP and two psychological measures (self-esteem and mood) | One-way ANOVA test | No significant differences in any of the measures between the groups before the interventions Reduced BP, increased self-esteem Rural and urban pleasant scenes effect on self-esteem than exercise-only control Green exercise has effects both in rural and urban Rural unpleasant scenes harm the benefits of exercise | Limitation: no exposure to real scenes of environment, while considering types of duration, intensity of physical activities. Green exercise has important health benefits |
| Song et al., 2018 [115] | Japan [Noda Hospital] Urbanicity not specified | 14 patients (Males, 4; females, 10) | Experimental study | 1min | 78.6 (±9.6) years | Nature viewing: Bonsai was used as visual stimuli | Bonsai has characteristic of mimicking natural landscapes and has been used in daily life in Japan since a longtime ago Japanese cypress bonsai trees The control group had no experimental stimulus | Viewing bonsai induces relaxation | Demographic factors: Age, gender Socioeconomic factors: N/A Comorbidity: N/A Environmental factors: N/A | Autonomic nervous activity HRV PR Prefrontal cortex activity | Paired t-tests were used to compare physiological responses between before and after viewing bonsai (pre- vs post-measurement) and between the two stimuli (bonsai vs. control) while Wilcoxon signed-rank test was used to compare psychological responses. | Increased parasympathetic nervous activity. Decreased sympathetic nervous activity Increased perceptions of feeling "comfortable" and "relaxed." | Limitation: studying psychological responses while viewing bonsai in healthy young people., small sample size, Viewing bonsai induces physiological and psychological relaxation. |
| Tsutsumi et al., 2017 [116] | Japan City not specified Urbanicity not specified | 12 healthy men | Experimental study | between February and March 2014 | 22.2 (±1.7 years) | Nature viewing Divided into two groups of 6 each and exposed to either sea or forest scenery by using the Visual Analogue Scale based on individual preference | Stimulation by viewing an individual's preferred video of sea or forest Watch 90 min DVDs of sea with natural sounds and forest with natural sounds No control groups. Two groups of six based on their preference for sea or forest scenery and each indicator was compared between them by using a pre post study design | Viewing an individual's chosen film of the sea or forest has an influence on relaxation. | Demographic factors: Male gender Socioeconomic factors: N/A Comorbidity: N/A Environmental factors: N/A | HRV Bispectrality Index System | Descriptive statistics Wilcoxon Signed-Rank test for the BP and POMS and the Mann–Whitney U-test for the HR, HF, and BIS were used | Differences in a decrease in HR, increase in high frequency, and sustained arousal level | Limitations: Healthy men in 20s, age range limited, no use of videos of personal preference, Viewing an individual's preferred video of sea or forest had a relaxation effect. Video relaxation therapy should be considered |

**Table 3.** (Continued)

| Author and year | Country, City, State Urbanicity setting | Sample Size | Study type | Follow up duration | Age (Mean ± SD, years) | Intervention Green space exposure | Exposure description + Greenspace type and Control group | Hypothesis/ Assumption | Covariates | CV related outcome | Statistical Analyses | Findings | Strengths & Weaknesses Conclusions |
|---|---|---|---|---|---|---|---|---|---|---|---|---|---|
| White et al., 2015 [117] | Southwestern England, UK Urbanicity not specified | 37 post-menopausal women | Experimental study | 1 week | 50.11 (±3.69) | Green exercise Cycling on a stationary exercise bike for 15 min while facing either a blank wall (Control) or while watching one of three videos: Urban (Grey), Countryside (Green), Coast (Blue). | Urban video: streets/ pedestrian walkways in a small town and featured shoppers, shops, and cars Green video: scenes of fields with sheep, hedgerows, and a small wood Blue video: headland overlooking a beach and of views from beach height across rocks and the sea Control group was the simulated urban "Grey" atmosphere | Simulated natural environment settings ("Green" and "Blue") to the neutral "Control" environment have any additional benefits to exercising in these settings beyond just exercising. | Demographic factors: Age, BMI Socioeconomic factors: N/A Comorbidity: N/A Environmental factors: N/A | SBP and DBP Valence and arousal | Repeated measures Analyses of Variance (ANOVAs) to examine the effect of Time of measurement and environment type for each variable | Outcomes were more positive in a simulated green and blue environment Blue environment led to shorter exercise duration and increases participants' willingness t to repeat it again in blue setting | Limitations: small number of environment types, simulated environment, no older women in sample size, Health benefits in natural environments More PA in natural environments |
| Bielinis et al., 2019 [118] | Olsztyn, Poland Urban | 21 | Experiment | 2 day | 23.86 ± 2.67 | Forest recreation–forest bathing | Forested area of the nature reserve No control groups. A pre and posttest design was employed | Two hypotheses: Participants' physiological and psychological relaxation can be influenced by a short-term forest leisure program. There is a utility of a forest near Olsztyn on the Redykajny nature reserve for forest leisure. | Demographic factors: Age, Gender, weight, height, BMI Socioeconomic factors: N/A Comorbidity: N/A Environmental factors: | Psychological measures PR, BP | Paired sample t-test was applied to compare pre-test and post-test Cohen's d was used to estimate the effect size | Negative mood markers were reduced after forest recreation; restoration and vitality increased PR, SBP and mean arterial pressures reduced after the program | Limitation: design was applied to one group, no control group, nervous system, and stress hormone levels were not assessed. Forest recreation lowers stress |
| Yu et al., 2017 [119] | Taiwan Xitou, central Taiwan Urbanicity not specified | 128 | Experimental study | 2 hours | 60.0 (± 7.44 years) | Forest bathing | Planted forest containing Cryptomeria japonica and Phyllostachys pubescents No control group. The study used a one-group pretest–posttest experimental design | There are physiological and psychological effects of a short forest bathing program on middle-aged and older people. | Demographic factors: Gender Socioeconomic factors: N/A Comorbidity: Diabetes, hypertension, heart diseases, other diseases Environmental factors: N/A | Physiological responses, PR, SBP, DBP, HRV, and psychological indices | Paired sample t-test was applied to compare pre-test measurements and post-test Cohen's d was used to estimate the effect size | Significant reduction in PR, SBP and DBP after the program No Significant change in HRV Forest bathing reduced mood states but vigor-activity increased Lowered anxiety levels | Limitations: Failure to collect information of confounding variables such as socio-economic status, medication usage, habits (e.g., smoking, exercise, etc.) and personality (e.g., nature lover); Environmental factors such as forest aesthetics, types and levels of pollutions and environmental conditions were not considered as covariates Short forest bathing program has health benefits, therapeutic properties and leads to relaxation |
| Koura et al., 2016 [120] | Japan City not specified Urbanicity not specified | 7 (5 females and 2 males) | Experimental design | 5–7 minutes | 76.2 (±6.7) | Horticultural therapeutic gardens | Walking in a horticultural therapeutic garden Not control group. The study used a pre post study design | There are benefits of horticulture therapy for all people's well-being that are reachable. | Demographic factors: Gender, age Socioeconomic factors: N/A Comorbidity: Dementia Environmental factors: N/A | HR variance Measures of sympathetic nervous system and parasympathetic nervous system | Not clear; Schematic view for data visualizations | The sympathetic nervous system (SNS: Low Frequency (LF)/ High Frequency (HF)) retracted while the parasympathetic nervous system (PNS: HF) was enhanced post interventions | Limitations not specified. Stress reduction effect of walking may last after the walk even among participants with moderate to severe dementia |

(*Continued*)

**Table 3.** (Continued)

| Author and year | Country, City, State Urbanicity setting | Sample Size | Study type | Follow up duration | Age (Mean ± SD, years) | Intervention Green space exposure | Exposure description type and Control group | Hypothesis/ Assumption | Covariates | CV related outcome | Statistical Analyses | Findings | Strengths & Weaknesses Conclusions |
|---|---|---|---|---|---|---|---|---|---|---|---|---|---|
| McEwan et al., 2021 [121] | United Kingdom City not specified Urbanicity not specified | 61 (50 females, 11 males) | Experimental design | 3 months | 18 years and older | Forest bathing | Three groups were used in 3X3 repeated measure experimental design: Forest bathing Compassionate Mind Training Forest Bathing combined with Compassionate Mind Training | Compassionate Mind Training [CMT] control condition would perform similarly to Forest Bathing. | Demographic factors: age, gender, height, sleeping and waking hours and use of medication Socioeconomic factors: smoking status; habitual alcohol consumption, weight, Comorbidity: N/A Environmental factors: N/A | Wellbeing and HRV | Multiple Analysis of Variance (MANOVA) and Cohen's d was used to estimate the effect size Independent t-tests were used to assess any differences in HRV scores between conditions at baseline | Positive emotions, mood disturbance, rumination, nature connection and compassion improved HRV increased | Limitations: Pragmatic constraints, sample size was limited by forest bathing sessions, women were only attracted to sessions, the biophobia was not considered, socioeconomically deprived individuals with need have no access to high quality greenspace, non-comparable HRV data from previous studies considered, effects of pandemic. Forest bathing has positive health effects and improves wellbeing |
| Park et al., 2017 [122] | Seoul, South Korea Urban | 21 women Gardening group (n = 11) Control group (n = 10) | Experimental study /Pilot study | 7.5 weeks | Gardening group 80.3 (±6.0) Control group 81.0 (±4.3) | Gardening intervention as a low to moderate PA intervention (green exercise) 15-session of gardening program (twice a week, average 50 minutes per session) from Sept. to Nov. 2015. | Planning a garden, making a garden plot, planting, sowing, mulching, fertilizing, watering, weeding, harvesting, garden maintenance, and cleaning the garden plot Exercise intensity of gardening intervention The control group matched on gardening intervention group was composed by the participants from the senior community center | Gardening intervention has impacts on blood vasculature, and immunity in women over 70 years old. | Demographic factors: Age (year); Height (cm) Body composition; Resting HR (beats/min); Education,; Elementary school graduate or less; Marital status, Socioeconomic factors: Income Comorbidity: Blood pressure; Cholesterol Antiarthritic Thyroid Heart disease Blood circulation Hip joints Osteoporosis Backache Environmental factors: N/A | Lipid profiles, BP, Pro-inflammatory proteins (TNF-α and Monocyte chemoattractant protein-1 (MCP-1) in peripheral-blood mononuclear cells (PBMC), and Oxidative stress markers: Inducible nitric oxide synthase (iNOS), Receptor for advanced glycation end products (RAGE) and the NADPH oxidase p47 | Chi-square tests were used to compare different variables Wilcoxon signed-rank test was used to compare before and after measurements | Gardening intervention as PA improves high density lipoprotein (LDL) profile, SBP and DBP and reduces oxidative stress Improved immunity in the intervention group Reduced TNF-α and RAGE No significant change for MCP-1, iNOS, and NADPH oxidase p47 | Limitations: small duration and small sample size. Gardening intervention has positive effects on lipid profiles, BP and therefore reduces the risk for CVD, improvement on some inflammatory markers (TNF-α) and oxidative stress (RAGE) of women aged over 70 years. |
| Song et al., 2013 [123] | Chiba, Japan Urban | 13 males | Experimental study | 15 minutes | 22.5 (± 3.1) years old | Urban parks (test) City area (control) | Urban green park Predetermined 15-minute walk sessions in an urban park (test) and in the city area (control) The control was the city areas around the urban park (city area) | Urban parks have similar health benefits to natural environments. | Demographic factors: Male gender, age Socioeconomic factors: N/A Comorbidity: N/A Environmental factors: N/A | HR and HRV Psychological responses | Paired t-test Wilcoxon signed-rank test | HR lower when walking in urban park than city Walking in the urban park enhanced the mood and decreased negative feelings and anxiety | Limitations: Female population not considered, age groups not considered, other ethnicities not considered, small sample size. Walking in urban parks has health benefits and relaxing effects in winter. |
| Wu et al, 2020 [124] | Hangzhou city, Zhejiang province, China Urban | 31 Control group (n = 11) Forest group (n = 20) | Experimental study/cohort study | 3 days | Control group: 73.91 (±6.6) Forest group: 73.50 (±5.9) | Forest bathing with Cinnamomum camphora (C. camphora) | C. camphora: Evergreen broad-leaved tree belonging to the family Lauraceae The control was a typical suburban area | Environments have varied effects on HTN patients. | Demographic factors: gender, age, body mass index (BMI) Socioeconomic factors: N/A Comorbidity: Hypertension, cardiac function class Environmental factors: N/A | BP, pulse oxygen saturation. HR, HRV. levels of plasma hsCRP Profile of mood states (POMS) | Categorical variables were compared by Chi-square analysis. Independent samples t-test or paired samples t-test was used to compare continuous outcomes | No significant differences at baseline across all variables DBP reduced in forest group Pulse oxygen saturation levels higher than control group Negative POMS was lower after forest bathing and there was a higher positive score. | Limitation: Sample size was small, elderly population, short intervention C. Camphora environment has good therapeutic effects on patients with HTN |

*(Continued)*

**Table 3.** (Continued)

| Author and year | Country, City, State Urbanicity setting | Sample Size | Study type | Follow up duration | Age (Mean ± SD, years) | Intervention Green space exposure | Exposure description + Greenspace type and Control group | Hypothesis/ Assumption | Covariates | CV related outcome | Statistical Analyses | Findings | Strengths & Weaknesses Conclusions |
|---|---|---|---|---|---|---|---|---|---|---|---|---|---|
| Lanki et al., 2017 [125] | Helsinki, Finland Urban | 36 (female) | Experimental study | 15-min period of sitting and viewing the environment, and a 30-min period of unhurried walking | 30–60 years | Green Exercise | Visit three different types of environments, namely: urban forest, urban park, and (built-up) city center. No control group: Before and after viewing measures were taken, when visiting environment types. Participants were considered as their own controls | Psychophysiological responses to visits to green areas are dependent on the quality of the area. | Demographic factors: age, female gender Socioeconomic factors: N/A Comorbidity: N/A Environmental factors: temperature, humidity, noise, respirable particles, pressure | SBP, DBP, HR and HRV were measured before and after the forest experience | Descriptive statistics Regression models | Visits to the green environments were associated with lower HR and higher HF than visits to city center. No differences in BP were observed between the green environments and city center | Limitations: No inclusion of both sexes. Even short visits to green areas may lead to beneficial changes in CV risk factors |

**Table 4. Characteristics for 5 studies with cancer-related outcomes.**

| Author and year | Country City State | Sample Size | Study type | Follow up duration | Age (Mean ± SD, years) | Intervention Green space exposure type + Hypothesis | Exposure description + Greenspace type and Control group | Hypothesis/ Assumption | Covariates | Statistical analyses | Cancer-related outcome | Findings | Strengths & Weaknesses Conclusions |
|---|---|---|---|---|---|---|---|---|---|---|---|---|---|
| Bail et al., 2018 [126] | Birmingham, Alabama, USA Urban | Total: 82 Intervention Group: 44 Control Group: 38 | Randomized controlled Trial study | 2 years | Total: 60.5 (± 9.4) Intervention Group 60 (± 8.4) Control Group 61 (± 10.5) | Mentored home based vegetable gardening | Raised bed/ grow boxes. Gardening supplies; gardening workbook, Master Gardener (MG) contact schedule. Contact information for their MG. Control group: BCS allocated to either 1 year vegetable gardening intervention or a wait list control group | There is a feasibility of a supervised home-based vegetable gardening intervention and health-related outcomes among breast cancer survivors (BCS) | Demographic factors: Age; Marital status; Body weight status. Socioeconomic factors: Current smoker; Race; Education; Rural county of residence; Currently employed; No. of individuals in household; Time since diagnosis; Functional limitations Comorbidities: Breast cancer stage; Cancer treatment; Comorbidities Environmental factors: N/A | Within-group comparisons over time were assessed using the paired t-test (interval data) and the McNemar test (dichotomous data) while baseline to post intervention change scores between groups were compared using the paired t test and the chi-square test | Health-related outcomes among breast cancer survivors (BCS): Vegetable consumption; PA; Health-related QoL; Physical performance; Anthropometrics and Biomarkers. | Compared with the control group, those in intervention indicated the enhancement in PA. The study reported the accrual, retention, and satisfaction Improved vegetable consumption, Continued gardening post two years | Limitations: modest sample size, no attention control group, one location area Feasibility of mentored, home-based vegetable gardening intervention Improvement in health behavior and outcomes among cancer survivors (BCS). |
| Blair et al., 2013 [127] | Alabama, USA Urban | 12 cancer survivors (eight adults, four children) | Feasibility study/ pilot study | 1 year | Adult survivors: 56.3 (± 4.4) Child survivors: 9.8 (± 1.0) | Vegetable gardening | Raised bed/ Earth boxes, receipt of soil mix, fertilizer, plants and seed gardening supplies. No control group: Post-intervention outcomes were compared to baseline, thus each participant considered as their own control. | Gardening increases fruit and vegetable intake, physical activity, quality of life, and physical functioning in cancer survivors, both children and adults. | Demographic factors: Age; Female gender Non-Hispanic white College education; BMI Socioeconomic factors: Ever smoker Servings/day fruit &vegetable; Days/week physical activity Comorbidity: Cancer treatment; Cancer type; Years since diagnosis Years since treatment completion; comorbid conditions Environmental factors: N/A | Descriptive statistics due to lack of power | Adult and child cancer survivors Assess the effects on fruit and vegetable intake, physical activity, quality-of-life, and physical function | Intervention well accepted and feasible among cancer survivors Improved strength, agility, and endurance among cancer survivors, Increased fruit and vegetable intake and PA. | Limitations: Small sample size, lack of control group, use pf self-report, Feasibility of gardening intervention Improved fruit and vegetables consumption, PA, and physical function in cancer survivors |
| Demark-Wahnefied et al., 2018 [128] | Alabama, USA Urban | 46 | Feasibility study/ Pilot Randomized Controlled Trial | 1 year | age 60+ | Seasonal vegetable gardens at survivors' homes | Plants, seeds, and gardening supplies Control group: Cancer survivors were assigned to a yearlong gardening intervention or a wait list-control arm | Home vegetable gardening can be feasible among older cancer survivors and is related to improved diet and other health related outcomes | Demographic factors: Age; Female sex, race, Education, Currently employed, Marital status, No. of people in household, Current smoker, Body mass index, Socioeconomic factors: Years since diagnosis. No. of functional limitations, Social readjustment events, Moderate to vigorous physical activity, Vegetable, and fruit intake Comorbidity: Type of cancer (Breast, prostate, Colorectal), Cancer treatment, No. of comorbidities Environmental factors: N/A | Paired t tests and McNemar's tests were used for within-group comparisons over time for interval and dichotomous variables, Paired t tests and Chi² tests were used for between-group comparisons of baseline to 1-yearfollow-up change scores. | Survivors of locoregionally-staged cancers Feasibility; accrual and retention; absence of serious adverse events and other outcomes and benefits | The retention, intervention, and appreciation of the trial Increases in reassurance of worth, waist circumference, Vegetable, and fruit consumption | Limitations: lack of statistical power, modest sample size, relying on self-reported data, the increased likelihood of Type I error associated with multiple comparisons. The feasibility of the study. Improved fruit and vegetables consumption, reassurance of worth and waist circumference Improve health, health behaviors and wellbeing of old cancer survivors |
| Li et al., 2008 [129] | Tokyo, Japan Urban | 12 males | Experimental study | 3 days | 45.1 (±6.7) | Forest bathing | Three-day/two-night trip to forest fields and to a city, in which activity levels during both trips were matched No control group: control measures were taken before and after the trips in working day. | There is an effect of a forest bathing trip on NK activity | Demographic factors: Age; male gender Socioeconomic factors: lifestyle habits of cigarette smoking, alcohol consumption, eating breakfast, sleeping hours, working hours, physical exercise, nutritional balance, and mental stress, Comorbidity:N/A Environmental factors: N/A | Two-way ANOVA with no-repeated measures One-way ANOVA with repeated measures Paired t-test Unpaired t-test | Natural killer cells (NK) activity, numbers of NK and T cells, and granulysin, perforin, and granzymes A/B expressing lymphocytes Adrenaline in urine | NK activity increased Increased numbers of NK, perforin, granulysin, and granzyme A/B expressing cells Decreased concentration of urine adrenaline NK activity increased and lasted for 7 days after forest trip No changes for all variables were observed for the city groups | Limitations not specified. Forest bathing trip has effect on health and the effect lasts for 7 days Phytoncides reduce stress and contribute partially to NK activity |

*(Continued)*

**Table 4.** (Continued)

| Author and year | Country City State | Sample Size | Study type | Follow up duration | Age (Mean ± SD, years) | Intervention Green space exposure type + Hypothesis | Exposure description + Greenspace type and Control group | Hypothesis/ Assumption | Covariates | Statistical analyses | Cancer-related outcome | Findings | Strengths & Weaknesses Conclusions |
|---|---|---|---|---|---|---|---|---|---|---|---|---|---|
| Li et al., 2007 [130] | Tokyo, Japan Urban | 12 males | Experimental study | 3 days | 43.1 (±6.1) | Forest bathing | Three-day/two-night trip to three different forest fields Blood prior to the trip was sampled as a control | There are effects of forest bathing on human NK activity. | Demographic factors: Age, male gender, Socioeconomic factors: lifestyle habits of cigarette smoking, alcohol consumption, eating breakfast, sleeping hours, working hours, physical exercise, nutritional balance, and mental stress. Comorbidity: N/A Environmental factors: N/A | Paired t-test | Natural killer (NK); NK cells, perforin, granzymes and granulysin-expression in peripheral blood lymphocytes (PBL). Proportions of NK, T cells, granulysin, perforin, and granzymes AlB-expressing cells in PBL | NK activity increased Increased NK, perforin, granulysin, and granzymes AlB-expressing cells | Limitations not specified. Forest bathing trip increase NK activity as a result of increasing the number of NK cells and induction of intracellular anti-cancer proteins |

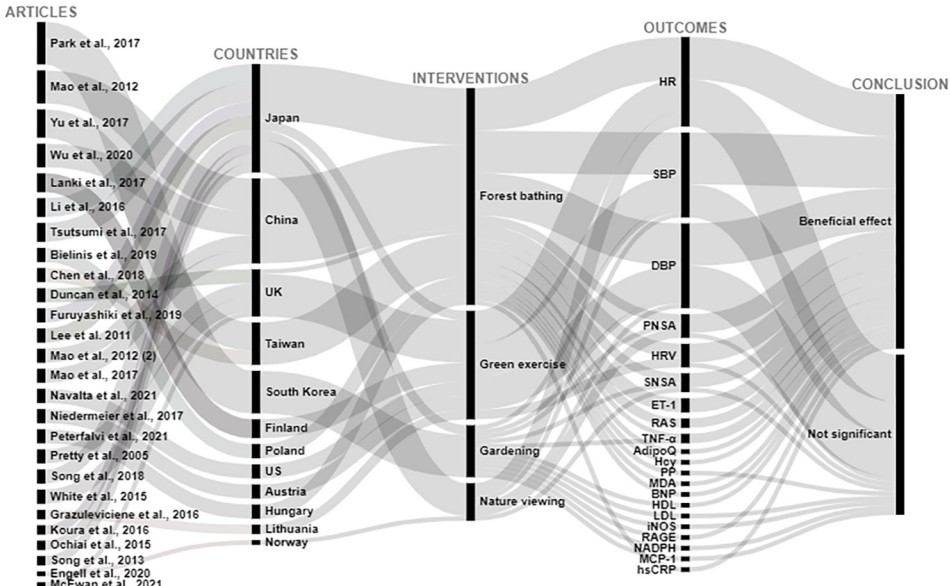

**Fig 2. Impact of greenspace or nature based interventions on CV health outcomes.** The first column represents articles, the second column represents geographical settings of studies, the third column represents specific interventions used in the studies, the fourth column represents the measures of CV health, while the fifth column represents the conclusion in terms of protective effects (*Beneficial effect*), or no significant results (*Not significant*). This graphical representation shows an overall trend in findings across all studies included. *Acronyms*: SBP[1]: Systolic blood pressure; DBP[2]: Diastolic blood pressure; BNP[3]: Brain natriuretic peptide; HRV[4]: Heart rate variability; RAS[5]: Renin-angiotensin system components; PNSA[6]: Parasympathetic Nervous System Activity; SNSA[7]: Sympathetic Nervous System Activity; hsCRP[8]: High sensitivity C-reactive protein; TNF- α[9]: Tumor necrosis factor alpha; HR[10]: Heart rate; MDA[11]: Malondialdehyde; RAGE[12]: Receptor for advanced glycation end products; iNOS[13]: Inducible nitric oxide synthase; MCP-1[14]: Monocyte chemoattractant protein-1; ET-1[15]: Endothelin-1; PP[16]: Pulse pressure; AdipoQ[17]: Adiponectin; Hcy[18]: Homocysteine; NADPH[19]: NADPH oxidase p47; HDL[20]: High-density lipoprotein; LDL[21], and Low-density lipoprotein.

and follow up after greenspace intervention. Two assessors discussed discrepancies between scores until a consensus was reached through a joint re-evaluation of the article, a method that has been used in previous studies [98]. The process resulted in a maximum of 9 possible points for each article; whereby 9 points represents the least risk of bias, and the risk of bias went up as the score went down. Following a cut-off point used in previous studies, score equal or greater than 5 was considered as "low-risk of bias" while score below 5 was considered as representing a high-risk of bias [99]. Our assessment suggested that 21 out of 31 studies (68%) had a low risk of bias; and the overall average score for all studies combined suggest a low risk of bias with a modified NOS score of mean (±SD) = 6.0 (±1.8). The risk of bias assessment is summarized in Table 2.

### 3.3. Summary characteristics of 31 articles included in the review

Data from included studies is summarized in two tables (3 and 4). Table 3 summarizes 26 studies with CV outcomes; and Table 4 summarizes 5 studies with cancer-related outcomes. Reported items include citation, study location, urbanicity setting, sample size, study type, follow up/duration, covariates, age, interventions, greenspace exposure type, CV health or cancer-related outcomes, statistical analyses conducted, main findings, study strengths and weaknesses, and conclusions.

### 3.4. Study design and demographics

All included studies used some kind of experimental designs. Thirteen (13) studies used simple pre-post study designs, some studies used the same group as the control and experimental group (on a different day/time) and measured statistical differences with paired sample t-tests [103,106,107,111,112,115,118,119,123,126,128–130], and eight (8) studies used randomized control and experimental groups and measured statistical differences with independent sample t-tests [100,104,105,108,109,113,124,129]. Sample sizes ranged from 7 [120] to 155 [104] with an average sample size of 33.5. Study participants' mean age ranged from 10 years [103] to 80.3 years [122]. Twenty (20) studies included both male and female participants, 7 included males only [106,107,112,116,123,129,130], and 4 included females only [111,117,122,125]. No study specified nonbinary gender conforming or transgender identity.

### 3.5. Statistical analyses

Various statistical approaches were used in describing data and testing effects of NBI on outcome measures of CV health and cancer-related QoL. Descriptive statistics reported means and standard deviations as well as frequency distributions [105,111,116,125,127]. In addition to descriptive statistics, inferential statistics were used to determine statistical differences observed pre and post intervention. Some studies used specific tests for normality and homogeneity of variances such as Kolmogorov-Smirnov and Levene's tests [100,108,109] or Shapiro-Wilk test [104,105]. Studies with normally distributed data used parametric tests such as t-tests, chi square, spearman correlation or regression [104]. Studies with categorical outcome variables used Chi squared test for statistical independence or association between samples [104,108,122,124,126,128]; and some studies with dichotomous outcome variables incorporated McNemar's test [126,128] to determine if there are differences between two related groups. Other studies used regression models to test predictions of interventions effects on dependent continuous outcomes variables [104,125]; and studies with more than two groups to compare during interventions used ANOVA to test for statistical differences between groups' means [101,103,110,114,117,121,129]. Other studies used post adjustment tests such as Bonferroni post-hoc pairwise comparisons or partial eta squared ($\eta$2) test [103] or Cohen's d test [118,119] for effect size estimation. In addition to parametric tests, studies with non-normally distributed outcome variables used nonparametric tests such as Mann Whitney U test for between subjects' comparisons or Wilcoxon Signed Rank tests for within subjects' comparisons to compare statistical differences between samples [100,102,116,122,123,104–106,108,109,112,113,115] or Kruskal-Wallis test for multi-group comparisons with post hoc Bonferroni adjustment [108]. One study used schematic views in representing their findings and did not specify the statistical test used [120].

### 3.6. Geographic distribution and urbanicity setting

Sixteen (16) studies were carried out in Asia, mostly in Japan and China, 12 in Europe, and 3 in North America. No study from other parts of the world (Africa, South America, and Australia) was identified. 22 studies were conducted in urban areas while 9 studies did not specify their urbanicity setting; and no study reported a rural setting for the experiment.

### 3.7. Summary of findings

Of 31 studies included in this review, 26 examined CV health related outcomes (Table 3) while 5 examined cancer-related outcomes (Table 4). Results of these studies are described separately for CV and cancer outcomes.

**3.7.1. Greenspace or NBIs on cardiovascular health.** Twenty-six (26) out of 31 studies included in the review looked at measures of CV health. Out of those 26 studies, 8 studies were conducted in Japan [104,106,107,112,115,116,120,123], 4 in China [100,108,109,124], 4 in the UK [103,114,117,121], and 2 in Taiwan [111,119]. One study was conducted in each of the following countries: Korea [122], Austria [110], Hungary [113], Poland [118], Lithuania [105], Finland [125], US [101] and Norway [102] (Fig 2). The most widely used intervention was forest bathing, quite common in Japan and China, followed by green exercise, nature viewing and gardening (Fig 2). The most reported outcomes were DBP, SBP, and HR, measured in 18 out of 26 studies. HRV was next and was measured in 5 out of those 26 studies, followed by measures of both the parasympathetic and sympathetic nervous systems, measured in 4 out of 26 studies. Few outcomes looked at stress measures of the cardiac myocyte such as the brain natriuretic peptide (BNP), Endothelin-1 (ET-1) and some components of the Renin-angiotensin system (RAS). Other outcomes investigated are measures of cholesterol such as high-density lipoprotein (HDL) and low-density lipoprotein (LDL). Most statistical tests conducted across all studies found that greenspace or NBI led to beneficial CV health outcomes (*Beneficial effect*), and some found no statistically significant difference (*Not significant*) (Fig *2*).

**3.7.2. Greenspace or nature-based interventions on cancer-related outcomes.** Five (5) out of 31 studies looked at cancer-related outcomes. Of those 5 studies, 3 were conducted in the US [126–128] while 2 were conducted in Japan [129,130]. Three US studies focused on vegetable gardening interventions while two Japanese studies focused on forest bathing interventions. Japanese studies looked at number of natural killer (NK) cells and their activity while US studies examined more diverse outcomes. Four of the outcome measures were related to positive health behaviors such as improved vegetable consumption habits [126–128], improved fruit consumption habits [127,128], increased PA [126,127] and gardening continuation [126]. Other outcomes were related to measures of physical fitness including strength [127], endurance [127], agility [127], and the two-minute-step test [126]. Three outcome measures were focused on overall health including weight loss [127], overall QoL [127,128], and reassurance of worth [128]. Three outcome measures were related biological markers including cortisol, a measure of stress [128], telomerase activity, a measure of aging [126,128], and interleukin-6 (IL-6), a pro-inflammatory biomarker and measure of systemic inflammation [128] (Fig 3). Observed trend suggests NBI's health protective effects on cancer outcomes (*Beneficial effect*) with few exceptional outcomes that were not statistically significant (*Not significant*) or significant only in control groups whereby control groups had better outcomes than the experimental groups (*Significant in controls*) (Fig 3). The 'significance in control groups' does not, in any way, suggest negative effect of the intervention. It is also not same as "*not significant*".

## 4. Discussion

### 4.1. Greenspace interventions and outcomes

This review focused on NBIs or greenspace interventions. Diverse types of experimental exposure to greenspace were identified, including forest bathing, green exercise, vegetable gardening, and nature viewing (Figs 2 and 3). Outcomes investigated were related to CV health or cancer. Study locations were distributed across three continents including Asia, Europe, and North America. As hypothesized, observed trends suggest overall beneficial effects of greenspace interventions on both CV health and cancer-related outcomes, with some exceptions on few outcome measures.

**4.1.1. Forest bathing.** Forest bathing "Shinrin-yoku" is a conscious and contemplative practice of being immersed in the sights, sounds, touches, tastes and smells of the forest [131]. This practice was developed in Japan in the 1980s as a physiological and psychological exercise

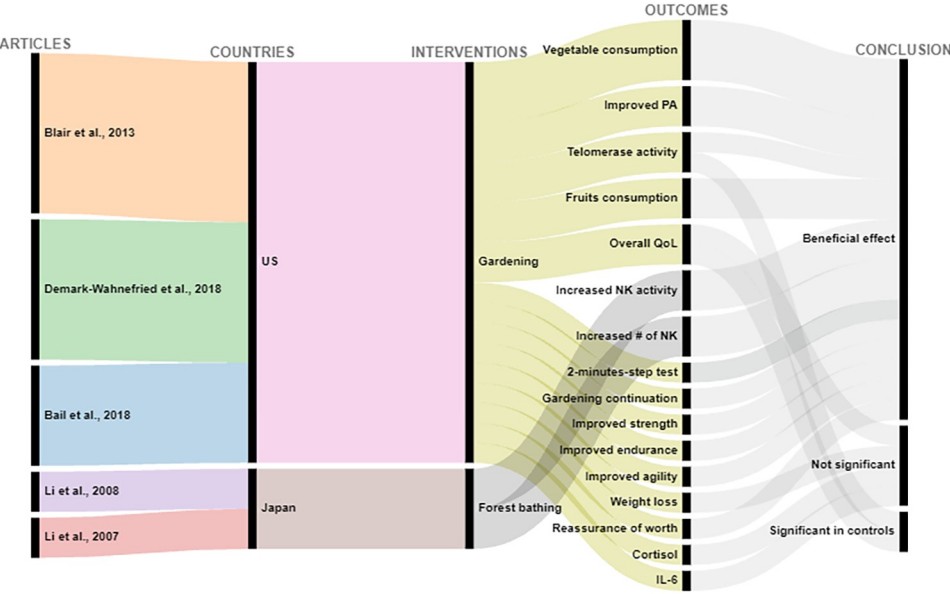

**Fig 3. The impact of greenspace or nature based interventions on cancer-related outcomes.** The first column represents articles, the second column represents geographical settings of studies, the third column represents specific interventions used in the studies, the fourth column represents the measures of cancer-related outcomes, while the fifth column represents the conclusion in terms of protective effects (beneficial effect), no significant results (not significant) or control groups had better outcomes than experimental groups (significant in controls).This graphical representation shows an overall trend in findings across studies. *Acronyms*: PA[1]: Physical activity; NK[2]: Natural killer cells; QoL[3]: Quality of life; and IL-6[4]: Interleukin-6.

and part of the national health program [132,133]. Its purpose was in twofold: (1) reduce burnout from the stressful work environment; and (2) inspire residents to reconnect with and protect the country's forests [133]. Scientists have then investigated its benefits on physical, mental, emotional, and social health outcomes [134]. Forest bathing is known to boost immunity [113,130,135], a plausible central pathway between nature exposure and human health benefits [136].

In this review, forest bathing was the most common intervention (15 out of 31 studies). Forest bathing was deployed in different forms including short forest recreation programs [113,118], forest therapy programs [111,112], longer slow walks in forests [100,104,107–109,124,129,130], forest viewing vs urban viewing [106], and full forest immersion experience, comprised of sessions of slowly moving in silence through woodland, stopping to observe using all of senses (sight, smell, touch, hearing, and taste) and engaging in slow and relaxing breathing to ensure discovery and mindful appreciation of the woodland [119,121]. In 15 studies with forest bathing intervention, 6 were conducted in Japan [104,106,107,112,129,130], 4 in China [100,108,109,124], 2 in Taiwan [111,119], and one in Hungary [113], Poland [118], and UK, respectively [121].

Most statistical tests conducted found beneficial effects of forest bathing on outcome measures for CV health with few exceptions that did not find statistically significant associations. Few non-significant associations included some outcome measures including diastolic blood pressure (DBP) [107,111,113,118], systolic blood pressure (SBP) [107,124], HR [104,109,111,113,124], pulse pressure [109], and HRV [119]. One study found no statistical significance in both PSNA and SNSA [119]. Other remaining statistical tests conducted across various studies found significant beneficial effects. The first beneficial outcome observed is in

measures of heart function such as reduced DBP [100,104,112,119,124], reduced SBP [100,104,111,112,118,119], lower HR [106,107,109,112,118,119], and increased HRV [121,124]. Another measured outcome that can impact CV health was stress. Stress reduction is salutogenic and was empirically observed with a decrease in stress hormones levels including urinary dopamine [107], adrenaline [112], and serum cortisol [109,112] after the intervention. Stress reduction was also observed with indicators of autonomic nervous system, such as enhanced parasympathetic nervous system activity (PNSA) [106] and suppressed sympathetic nervous system activity (SNSA) [106].

Improved systemic inflammatory profile is another beneficial outcome that was observed through reduction in both pro-inflammatory biomarkers and increase in anti-inflammatory biomarkers after forest bathing interventions. Reduced pro-inflammatory biomarkers include endothelin (ET-1) [100,108,109], IL-6 [100,108,109], tumor necrosis factor alpha (TNF-α) [109], homocysteine (Hcy) [100], and high sensitivity C-reactive protein (hsCRP) [124]. Increased anti-inflammatory biomarkers include serum adiponectin [107]. There were numerical differences between pre and post measures for two measured biomarkers of inflammation within the intervention groups, but no statistically significant differences were observed. Those non-statistically significant tests were for TNF-α [100,108] and hsCRP [108], and were reported in the alluvial chat as "Not significant".

Measures of oxidative stress were also improved after forest bathing interventions, as observed through lower levels of malondialdehyde (MDA) in experimental group post-intervention [108,109]. Last but not least, measured CVD pathological factors biomarkers were improved after forest bathing interventions as observed though serum reduction of constituents of the renin angiotensin system (RAS) (renin [108], angiotensin II (Ang II) [108], angiotensinogen (AGT) [100,108], angiotensin II type 1 receptor (AT1) [100,108], and angiotensin II type 2 receptor (AT2) [100,108]) and the brain natriuretic peptide (BNP), a biomarker of HF [108]. One study found mild reduction in renin and angiotensin II (Ang II) in the experimental group, although changes were not statistically significant [100]; and this was reported as "Not significant" in alluvial charts.

Most statistical tests conducted found beneficial effects of forest bathing on cancer-related measured outcomes including enhanced immune functioning observed through increase in number of NK cells [129,130] and their activity [129,130]. The forest bathing 'outcome-conclusion' chart is illustrated in Fig 4.

Forest bathing is a promising intervention to improve CV health and QoL, particularly during cancer survivorship. Clinical practitioners, particularly those working in cardio-oncology specialties should examine more closely these non-invasive interventions and incorporate them in the standard of care to optimize CV health outcomes for cancer survivors through increased use of nature prescription programs, in addition to the clinical standards of care.

**4.1.2. Green exercise.** Another commonly used intervention was green exercise (8 out of 31 studies). Green exercise has been defined as any PA occurring in a natural environment [114]. In this study, exercising with a view of nature through a window, on pictures, or on televisions was also considered "green exercise". Diverse green exercise interventions were used in studies included in this review, but most of them used nature visual stimuli. Duncan et al., 2014 had participants in the intervention arm of their study cycle for 15 min whilst watching a film of cycling in a forest environment [103]. Like Duncan et al., 2014, Pretty et al., 2005, had participants watch different scenes of videos projected on a wall whilst exercising on a treadmill [114] while Song et al., 2018's participants viewed Bonsai, small plants in container with restriction to roots or food storage capability [137]. The Bonsai used as a visual stimulus had characteristic mimicking natural landscapes that has been historically used in daily life in Japan [115]. White et al., 2015 also had their participants in the intervention arm cycle on a

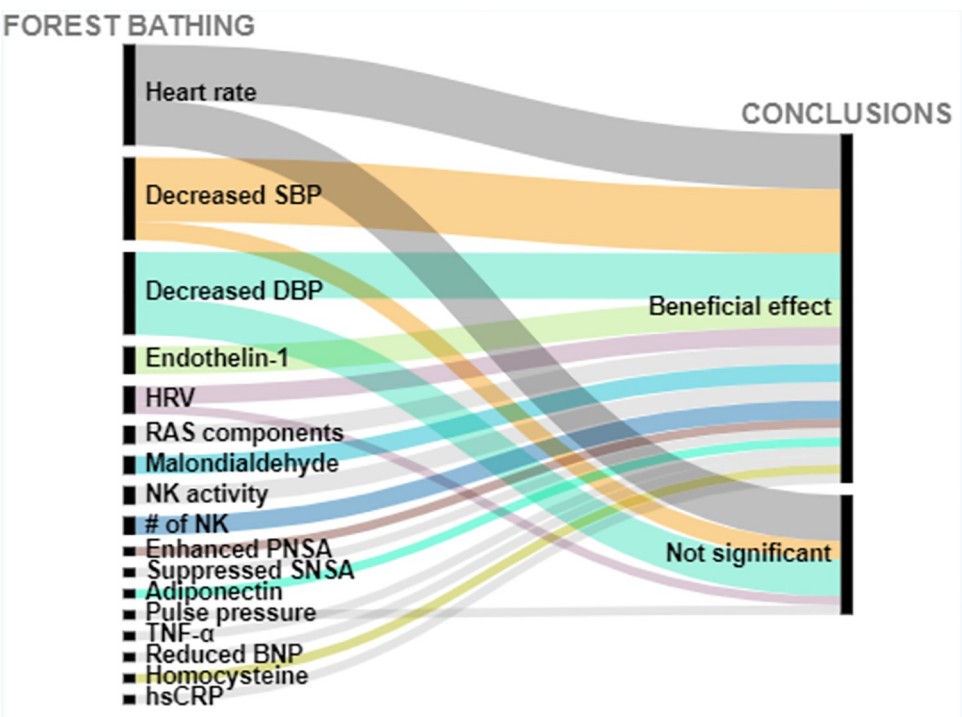

**Fig 4. Forest bathing intervention effects on both CV health and cancer-related outcomes: Trends of associations among all statistical tests conducted.** The first column represents outcome measures while the second column represents summary conclusions in terms of protective effects (beneficial effect) or no significant results (not significant). *Acronyms*: SBP[1]: Systolic blood pressure; DBP[2]: Diastolic blood pressure; BNP[3]: Brain natriuretic peptide; HRV[4]: Heart rate variability; RAS[5]: Renin-angiotensin system components; PNSA[6]: Parasympathetic Nervous System Activity; SNSA[7]: Sympathetic Nervous System Activity; hsCRP[8]: High sensitivity C-reactive protein, TNF- α[9]: Tumor necrosis factor alpha; and NK[10]: Natural killer cells.

stationary exercise bike for 15 min while watching one of three videos: Urban (Grey), Country-side (Green), or Coast (Blue) [117]. Grazuleviciene et al., 2016 had the participants in intervention arm of their experiment walk in a pine forest park [105]. Lanki et al., 2017's intervention consisted of a visit to an urban greenspace (forest or park) [125] with 15 min visit of sedentary viewing greenspace and 30 min of walking in greenspace [125]. Navalta et al., 2021 used exercise in a desert environment (brown environment) as a nature-based intervention to test if there are similar benefits to those anticipated in green environments [101]. Niedermeier et. al., 2017 used mountain hiking as green exercise [110].

Green exercise has been shown to improve both physical and mental health [114,138] and higher enjoyment of exercise [139]. Some of the positive health outcomes previously associated with green exercise include greater feelings of revitalization and positive engagement [140], and improvement in measures of mood and self-esteem [141] such as depression, tension, and anger [142,143]. Green exercise has been suggested by previous scholars as a potential work-place intervention to reduce job stress and promote restoration [144]. Chronic stress has been linked to increased CVD risk [145–147], including a 40–50% increase in the occurrence of coronary heart disease in prospective observational studies [145,148].

In our review, interventions with green exercise were conducted in different countries, including the UK [103,114,117], Lithuania [105], Finland [125], Austria [110], Japan [123] and the US [101]. Green exercise was found to be positively associated with many outcome measures related to CV health with few statistical tests that found no significant associations or no

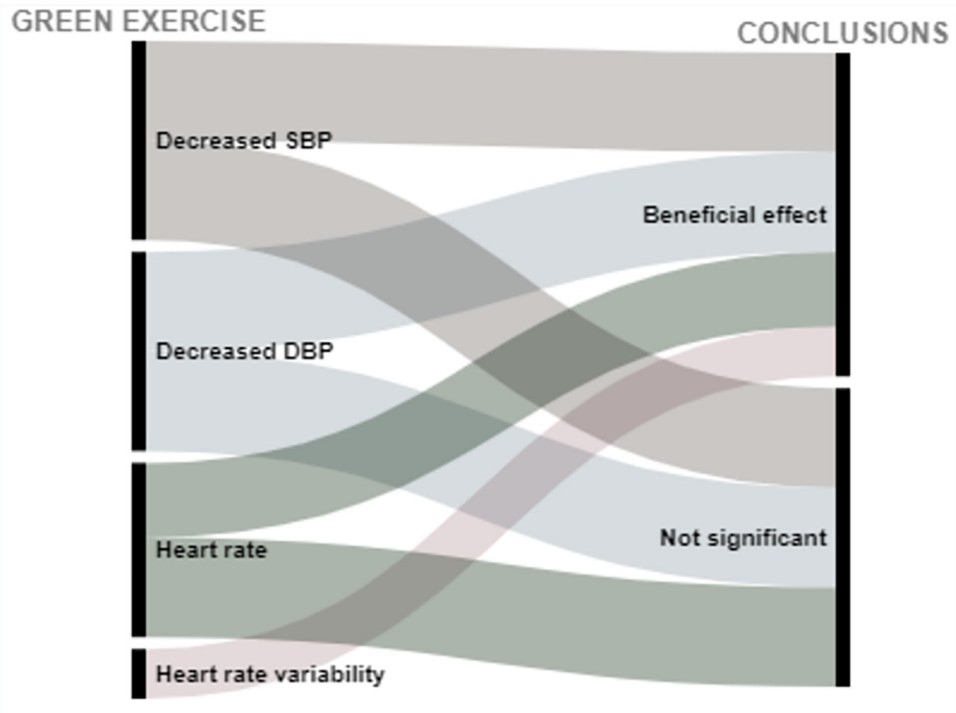

**Fig 5. Green exercise intervention effects on CV health outcomes: Trends of associations among all statistical tests conducted.** The first column represents outcome measures while the second column represents summary conclusions in terms of protective effects (beneficial effect) or no significant results (not significant). *Acronyms*: SBP[1]: Systolic blood pressure and DBP[2]: Diastolic blood pressure.

numerical difference at all. Green exercise's beneficial CV health outcomes include observed reduction in SBP [103,114,117], DBP [105,114,117], HR [117,123,125], and increase in HRV [123,125]. Another significant change observed was a reduction in cortisol, a measure of stress [105]. Some studies did not find a significant difference on measures of SBP [101,105,110,125], DBP [101,103,110,125], and HR [101,110,114], including one study that found no association between green exercise and one measure of CV health, HR [103]. For studies that looked at cancer-related outcomes, none used a green exercise intervention. The green exercise 'outcomes-conclusions' chart is illustrated in Fig 5.

No study in this review investigated the impact of "green exercise" on cardiotoxicity among cancer survivors. This literature gap suggests the need for empirical investigation on the role of greenspaces in reducing risks for cardiotoxicity in this highly vulnerable population and testing the use of such interventions in Cardio-oncology clinics to optimize CV health and improve cancer survivorship care. Additionally, only one statistical test investigated the impact of green exercise on CV biological markers by looking at cortisol. Future studies should investigate more biomarkers, including additional stress biomarkers and CVD pathological factors such as the components of the renin angiotensin system and inflammatory biomarkers such as IL-6, hsCRP, TNF-α etc..

**4.1.3. Vegetable gardening.** Gardening interventions provide individuals with hands-on experience planting, growing, and harvesting fruits and vegetables, which may promote consumption of fruits and vegetables [149,150]. Individual benefits of gardening activities include increased PA, access to fresh air, landscape beautification and enjoyment [151]. Gardening interventions have been linked to many health benefits [152] including improved physical

health [153,154] and mental health [155–157]. Gardening has been proposed as a strategy for health promotion in aging women [158] and its prescription, along with other conservation activities are recommended to improve health and wellbeing in aging population [159]. In the cancer care continuum, gardening interventions have been linked to positive health outcomes and improved survival [160,161]. Some specific benefits of gardening during cancer survivorship include improved dietary habits, improved PA, and improved QoL [162].

In this review, vegetable gardening interventions were conducted in Japan [120], South Korea [122] and the US [126–128]. Two studies looked at CV health related outcomes [120,122] while three studies looked at cancer-related outcomes [126–128]. Most studies found beneficial effects of gardening interventions on outcome measures related to CV health and cancer-related QoL, with some exceptions that found no statistically significant changes (not significant), or significant only among controls. Those 'not significant' exceptions include some statistical tests on outcome measures of weight loss and overall QoL in one feasibility study in cancer survivors [127], some biomarkers including the monocyte chemoattractant protein-1 (MCP-1), NADPH oxidase p47, and the inducible nitric oxide synthase protein (iNOS) [122], stress hormone cortisol and IL-6 [128], and low-density lipoprotein (LDL) [122]. Two tests found significance among controls, one on overall QoL [128] and another one on telomerase activity [126], an enzyme responsible for maintenance of telomeres length by addition of guanine-rich repetitive sequences in both gametes and stem and tumor cells [163].

Included studies in this review showed beneficial effects of gardening interventions on stress [120], total cholesterol and HDL [122], BP [122], dietary habits [126–128], positive self-care behaviors [126,127], physical performance [127], increased reassurance of worth [128] and improved aging process [128]. Stress reduction benefits were observed through proxy measures with enhanced parasympathetic nervous system activity (PNSA) [120] and suppression of sympathetic nervous system activity (SNSA) [120]. Benefits on blood cholesterol level were measured through improved high-density lipoprotein [HDL], or good cholesterol profile [122]. Beneficial outcomes in BP were measured with both decreased SBP [122] and decreased DBP [122]. Improvement in dietary habits was observed through improved vegetable and fruit consumption [126–128]. Positive self-care behaviors were observed through improved PA [126,127] and gardening continuation [126]. Physical performance improvement was observed through improvement in the 2-minute-step test [126] and other measures including improved strength [127], improved endurance [127] and improved agility [127]. Increased reassurance of worth was measured with self-reported assessments of psychosocial measures [128]. Improvement in aging process was observed through a decrease in telomerase activity [128]. The gardening interventions impact on both CV health and cancer-related outcomes, along with the overall conclusion are graphically illustrated in Fig 6.

Observed trends (Fig 6) suggest that gardening is a promising intervention to improve outcomes related to CV health and QoL during cancer survivorship. Cardio-oncologists should keep close collaborations with primary care providers in optimizing the cancer survivorship care by including these innovative interventions to improve CV health and survivorship experience. Community leaders, including local government and other community-based organizations should work together to ensure presence, accessibility, and use of community gardens. In addition to supporting positive healthy gardening behaviors, those gardens also have potential to increase access to healthier foods options for residents in "food deserts" and "food swamps" neighborhoods [164–166]. Such gardens could also enhance biodiversity, local ecosystem, water management and contribute to local climate change resilience strategies [167]. Additionally, continuous targeted messaging campaigns should be in place to remind those at increased risk of the benefits associated with gardening. Academic partners should come in to continuously evaluate impact and suggest best practices to ensure maximum benefits from all

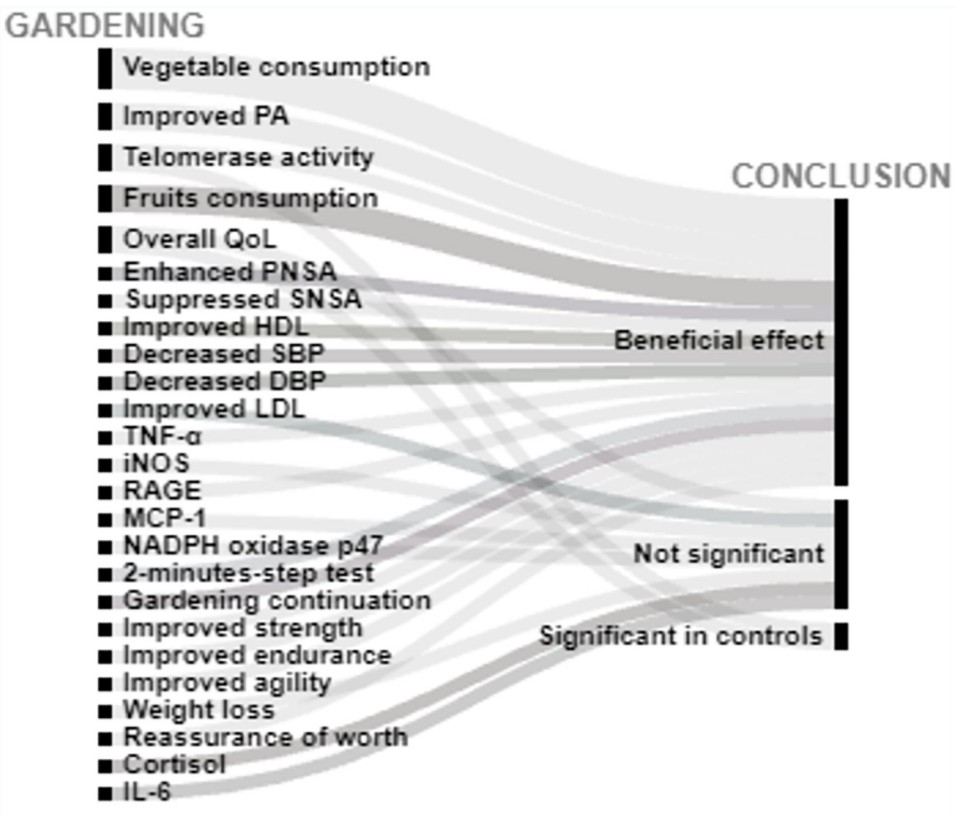

**Fig 6. Vegetable gardening intervention effects on outcomes related to CV health and cancer: Trends of associations among all statistical tests conducted.** The first column represents outcome measures while the second column represents summary conclusions in terms of protective effects (beneficial effect), no significant results (not significant) or control groups had better outcomes than experimental groups (significant in controls). *Acronyms*: PA[1]: Physical activity; QoL[2]: Quality of life; PNSA[3]: Parasympathetic Nervous System Activity; SNSA[4]: Sympathetic Nervous System Activity; SBP[5]: Systolic blood pressure; DBP[6]: Diastolic blood pressure; HDL[7]: High-density lipoprotein; LDL[8]: Low-density lipoprotein; TNF- α[9]: Tumor necrosis factor alpha; RAGE[10]: Receptor for advanced glycation end products; iNOS[11]: Inducible nitric oxide synthase; MCP-1[12]: Monocyte chemoattractant protein-1, and IL-6[13]: Interleukin-6.

the resources set aside for such a community wide intervention to support intergeneration equity. Future studies should incorporate more biological measures including pathological factors for CVD such as biomarkers of oxidative stress and more inflammatory biomarkers in addition to IL-6, the only pro-inflammatory biomarker that was investigated in vegetable gardening interventions studies included.

**4.1.4. Nature Viewing.** Exposure to natural environments including viewing them has been linked with improved restoration and cognitive capacity [102] and autonomic function recovery after acute-mental stress [168]. In this review, we found studies that tested nature viewing effects on measures of CV health including HR, SBP, DBP, PSNA and SNSA. Those studies were carried out in two countries, Japan [115,116] and Norway [102]. Statistical tests found beneficial effects of nature viewing on CV health including reduction in HR [102,116], enhanced PSNA [115,116] and suppressed SNSA [115]. Tests on measures of SBP and DBP were not statistically significant [116]. The nature viewing interventions on both CV health outcomes, along with the overall conclusion are graphically illustrated in Fig 7.

Contrary to other NBI in this review (forest bathing, green exercise, and vegetable gardening), nature viewing intervention did not measure a single biological marker of inflammation.

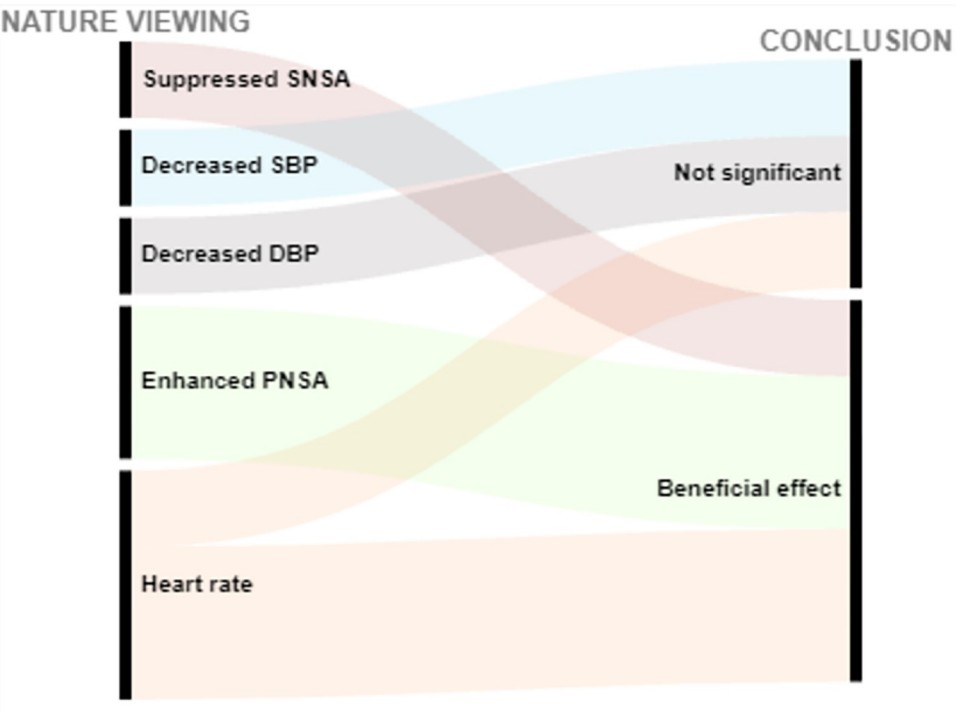

**Fig 7. Nature viewing intervention effects on CV health outcomes: Trends of associations among all statistical tests conducted.** The first column represents outcome measures while the second column represents summary conclusions in terms of protective effects (beneficial effect) or no significant results (not significant). *Acronyms*: PNSA[1]: Parasympathetic Nervous System Activity; SNSA[2]: Sympathetic Nervous System Activity; SBP[3]: Systolic blood pressure and DBP[4]: Diastolic blood pressure.

Future studies should investigate nature viewing's impact on biomarkers including CVD pathological factors such as the components of the renin angiotensin system and pro-inflammatory biomarkers such as IL-6, hsCRP, and TNF- α. Such knowledge would complement current behavioral self-care plans particularly for cancer survivors in reducing risk for cardiotoxicity; and nature viewing is a relatively harmless intervention, amenable to change and relatively easier to implement.

Presented all together, this review suggests that forest bathing and gardening interventions have the most beneficial outcomes (Figs 4 and 6) compared to other interventions (nature viewing or green exercising) which are also beneficial, but to a less extent (Figs 5 and 7). Intervention specific alluvial charts suggest more thickness for "beneficial effect" for forest bathing and gardening compared to nature viewing or green exercise. These findings have implications for increasing use of forest bathing and/or gardening interventions to improve CVD and/or cancer outcomes. Nature viewing and green exercise interventions remain also very important in improving outcomes. The clinical use of these interventions would be best assessed with patient preference and what interventions they are most likely to adhere to.

## 4.2. Limitations

While this review is methodologically rigorous, it has some limitations. First, in the risk of bias assessment, we used a modified version of the Newcastle Ottawa Scale because there was no validated tool that accurately assessed all types of studies included in our review. While the official NOS has been validated for case-control and cohort studies, the scoring guide created

for this study by modifying the scale to capture factors related to experimental or pre-post studies has not been validated, and it's scoring can be subjective. This subjectivity was attenuated by ensuring that two reviewers (J.C.B. and J.S.B.) independently assess all studies. Secondly, we reported trends across all relevant statistical tests conducted in all included studies with alluvial charts to visualize our results summary, but no meta-analysis was done to suggest any statistical inference for all the articles if taken altogether. Therefore, our trend across all studies should be seen as a descriptive summary of findings; and any inference made should consider all studies collectively. Thirdly, not all included studies measured, adjusted for, or reported the same variables. This is why we used alluvial chart to summarize similar trends instead of conducting a meta-analysis to deduct any statistical inference for all studies combined. Last, this review is not immune to other limitations discussed by the authors of the included studies, which may include small sample size, lab errors, potential misclassification, or other measurement errors. Regardless of the limitations, this review is an outstanding summary of impact of greenspace or nature based interventions on both CV health and cancer-related outcomes and highlight benefits with direct implication for clinical and public health practice.

## 5. Conclusion

This review sought to assess the impact of greenspace or NBI on: (1) CV health, and (2) cancer-related outcomes.

Interventions used included a Japanese tradition of forest bathing or "shinrin-yoku," green exercise, gardening, and nature viewing. CV health related outcomes include measures of BP, HR, HRV, autonomic nervous system activity, stress biomarkers including cortisol, oxidative stress measures such as iNOS, RAGE, and NADPH oxidase p47, CVD pathological factors including lipid profile, components of the renin angiotensin system, pro-inflammation biomarkers including IL-6, hsCRP, TNF- α, ET-1, Hcy, MDA, and MCP-1 and anti-inflammatory biomarkers including adiponectin. Cancer-related outcome measures include measures of physical performance such as physical strength, endurance, and agility; personal behaviors such as vegetable and fruits consumption, PA, and weight loss; biological markers including stress markers (cortisol), inflammatory markers (IL-6), some components of the renin angiotensin system (RAS), and some immune function markers including both the count of natural killer cells as well as their activity.

An overall trend across studies suggests beneficial effects of greening and NBI on both CV health and cancer-related outcomes, although not all studies found a significant benefit. Cardio-oncologists, along with primary care providers should incorporate these innovative interventions in the standard of care to optimize both CV and cancer-related health outcomes.

Future studies should combine multiple measures of CVD pathological factors including components of the renin angiotensin system (renin, Ang II, AGT, AT1 and AT2), multiple markers of oxidative stress, multiple measures of both pro and anti-inflammatory biomarkers, and multiple biomarkers of stress. Other direct and relatively easier measures such as BP, HR, pulse pressure and HRV would be important to add to this line of investigation. Additionally, future studies should pay more attention to some populations with higher CVD risk such as cancer survivors to order to investigate the premise of such innovative population-based approaches in reducing cardiotoxicity from cancer treatment therapies and optimize the survivorship experience.

Existing conceptual models such as the "*Greenspace and Health Equity model*" [169] or the "*Greenspace in Cardio-Oncology model*" [1] can be very useful in future research on greenspace and CardioOncology disparities. There is a need for increased research funding from relevant

organizations such as the American Heart Association, the American Cancer Society, and National Health Institutes including the National Cancer Institute. This knowledge will promote a more robust understanding of the role of greenspace and NBI on CV and/or cancer-related outcomes as well as their critical contribution to climate resilient neighborhoods. The focus on biomarkers is particularly relevant for clinical practice as more biomarkers can clinically be measured and greenspace interventions impact on CV health can be continuously assessed during all stages of the cancer care continuum. Such practice can help reduce risks for MACE, reduce mortality, and improve cancer survivorship quality and survival.

## Supporting information

**S1 Checklist. The PRISMA 2020 checklist: Appendix A.**
(DOCX)

**S1 File. The full databases search strategy and alluvial charts data files: Appendices B and C.**
(DOCX)

## Acknowledgments

Thank you to Rita Sieracki, at the Medical College of Wisconsin Library for the help in leading the systematic database search.

## Author Contributions

**Conceptualization:** Jean C. Bikomeye, Kirsten M. M. Beyer.

**Data curation:** Jean C. Bikomeye, Joanna S. Balza, Kirsten M. M. Beyer.

**Formal analysis:** Jean C. Bikomeye.

**Funding acquisition:** Kirsten M. M. Beyer.

**Investigation:** Jean C. Bikomeye, Kirsten M. M. Beyer.

**Methodology:** Jean C. Bikomeye, Joanna S. Balza, Jamila L. Kwarteng, Andreas M. Beyer, Kirsten M. M. Beyer.

**Project administration:** Jean C. Bikomeye, Joanna S. Balza, Kirsten M. M. Beyer.

**Resources:** Jean C. Bikomeye, Jamila L. Kwarteng, Andreas M. Beyer, Kirsten M. M. Beyer.

**Software:** Jean C. Bikomeye, Joanna S. Balza.

**Supervision:** Kirsten M. M. Beyer.

**Validation:** Jean C. Bikomeye, Jamila L. Kwarteng, Andreas M. Beyer, Kirsten M. M. Beyer.

**Visualization:** Jean C. Bikomeye.

**Writing – original draft:** Jean C. Bikomeye, Kirsten M. M. Beyer.

**Writing – review & editing:** Jean C. Bikomeye, Joanna S. Balza, Jamila L. Kwarteng, Andreas M. Beyer, Kirsten M. M. Beyer.

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
