## [Decision Letter · Decision Letter 0]

31 May 2022

PONE-D-22-07911The Impact of greenspace or nature-based interventions on cardiovascular health or cancer related outcomes: A systematic review of experimental studiesPLOS ONE

Dear Dr. BIKOMEYE,

Thank you for submitting your manuscript to PLOS ONE. After careful consideration, we feel that it has merit but does not fully meet PLOS ONE’s publication criteria as it currently stands. Therefore, we invite you to submit a revised version of the manuscript that addresses the points raised during the review process.

Sorry for the delay in reviewing. However, we have now received two reviews. One reviewer has highlighted major revisions and having read the manuscript and gone through the suggestions, I think this is appropriate. Please can you address the concerns raised in particular around the differences between the PROSPERO protocol and the methods.  Reviewer 2 should have put minor revisions and has listed a few things to correct. The paper is well written and has interesting data, therefore once the revisions have been addressed it should be a worthwhile addition to the research topic.

We look forward to receiving your revised manuscript.

Kind regards,

Lindsay Bottoms

Academic Editor

PLOS ONE

Journal Requirements:

“The work was supported by an American Heart Association Scientific focused research network on disparities in Cardio-oncology (K.M.M.B. and A.M.B.) grant, NIH (National Institutes of Health) grants: R01HL133029 (A.M.B.), R01CA214805 (K.M.M.B), the Medical College of Wisconsin Cancer Center grants (KM.M.B), and by the We Care Fund (A.M.B.).”

Reviewers' comments:

Reviewer's Responses to Questions

**Comments to the Author**

1. Is the manuscript technically sound, and do the data support the conclusions?

Reviewer #1: Partly

Reviewer #2: Yes

2. Has the statistical analysis been performed appropriately and rigorously? 

Reviewer #1: N/A

Reviewer #2: N/A

3. Have the authors made all data underlying the findings in their manuscript fully available?

Reviewer #1: Yes

Reviewer #2: Yes

4. Is the manuscript presented in an intelligible fashion and written in standard English?

Reviewer #1: Yes

Reviewer #2: Yes

5. Review Comments to the Author

Reviewer #1: The manuscript presents a systematic review of the impact of green spaces and nature-based interventions (NBIs) on cardiovascular and cancer outcomes. The review included 31 experimental studies which indicate a potential beneficial health effect of some form of NBI, although the available evidence presents considerable heterogeneity. The review protocol is registered in PROSPERO and the PRISMA guidelines have been adhered to, with some minor inaccuracies.

While the search appears to have been done rigorously, there are some issues that require addressing, as listed below:

1) Introduction - Sections discussing greenspace would benefit from presenting a definition of this. "Greenspace" refers in the literature to many different things (e.g., urban parks, wild nature, gardens, normalized difference vegetative index) and may at times include blue spaces. What definition, if any, guided this systematic review?

2) Introduction - The review includes nature-based interventions as part of the search, but these are not discussed in the Introduction with regards to their definition and rationale for including. What is the added value of looking at NBIs as well as greenspaces? Where do NBIs sit within the MBASIC framework?

3) Introduction - A recent systematic review with meta-analysis looked at the relationship between green spaces and CVD (https://doi.org/10.1016/j.envpol.2022.118990). What does this review add?

4) Section 2.2 Article selection process- The criteria for intervention should be more clearly described. Any type of green space? Does the "other" means here that you looked at interventions not taking place in green spaces?

5) Section 2.2 Article selection process - The criteria for cancer related outcomes described here (lifestyle changes and QoL) do not seem to match the outcomes described in the Introduction as well as section 2.1 (which include cancer prognosis, cancer incidence, cancer mortality, etc.). I would encourage the authors to clarify what cancer-related outcomes were investigated and provide a rationale for looking at these outcomes.

6) Section 2.3 Eligibility Criteria - Table 1 presents as inclusion criterion "empirical studies" but this does not fully reflect the specific designs that appears to have been included in the review, i.e., experimental with or without control and quasi-experimental. Please clarify

7) Section 2.4 Data extraction - The information presented here does not match the PROSPERO protocol, which is much more comprehensive than the one presented here (see extract from protocol below). I also note that covariates are not discussed in the review, despite this was included in the protocol. Please clarify why data extraction did not adhere to the protocol.

From the PROSPERO protocol: "We will extract the following data from articles: (1) Studies geographical information (City, state, country); (2) Studies urbanicity setting (rural, semi-urban, or urban); (3) The type of greenspace or nature-based interventions + assumptions made or hypotheses; (4) Measures of any CVD related outcome (Incidence, morbidity, or CVD related mortality); (5) Measures of any cancer related outcome (anything from the cancer control continuum, cancer related quality of life (QOL), or cancer related mortality; (6) CVD or cancer type under investigation (specific or any type); (7) All covariates adjusted for: a. Individual level factors: i. Demographic information (when available); ii. Socioeconomic information (when available); iii. Co-morbidity information (when available). b. Neighborhood factors (when available): i. Social environment factors, ii. Other neighborhood-built, environment characteristics. (8) Statistical analyses conducted; (9) Studies strengths and weaknesses"

8) PRISMA flowchart: In the screening phase, please clarify the criteria for excluding 45 abstracts.

9) Risk of bias: The authors appears to have used the NOS for cohort studies, but given the inclusion of experimental studies only, it is unclear why a more appropriate tool, like the ROB2 or ROBINS-I, was not used. The authors discuss this in the limitations, but the reason for not using other tools beyond NOS should be clarified.

Importantly, the criterion used for assessing the representativeness of the exposed group does not match the criterion set in the NOS scale. Here, representativeness should related to the community where the study took place. Judging representativeness based on whether the sample used in the study matches the title/abstract can be prone to bias per se. I would strongly encourage the authors to reassess this criterion and provide a clearer justification for the choice of tool to assess risk of bias.

10) Section 3.4 Study design and demographics - Were the samples included in the studies composed of healthy individuals or patients populations? While this may be self-evident for the cancer studies, it is unclear for CVD and it would be useful to tease out whether the interventions work as a preventative measure (i.e., maintaining good CVH among healthy individuals) or remedy (improving outcomes among people with CVD).

11) Sections 3.7.1 and 3.7.2 - Two main comments here: 1) The types of activities completed by the control groups (if present) should be discussed, as it is unclear what the NBIs were compared to; this should also be evident in Table 3 and Table 4, which should indicate whether a control group was included and what they did. 2) It is unclear how the reviewers decided on a beneficial vs. nonsignificant effect. This requires better clarification. Were effect sizes considered for this? Also, did any of the studies find a negative effect of green space or NBIs?

12) Figure 2: I acknowledge the effort made by the authors in this visual depiction, but I must admit that it requires quite a lot of effort to make sense of. For instance, it is unclear what criterion determined the thickness of each study. Should the studies be organised based on region of the world or continent rather than country? Could a pattern or colour code like in Figure 3 be used to distinguish beneficial effect from "not significant"?

13) Section 3.7.2 and Figure 3 - What is defined here as "significant in control only" deserves better clarification. Does this imply inferiority of the NBI compared to the control, i.e., a negative effect of the intervention? Or is this related to no change observed in the intervention?

14) Discussion - This section would benefit from a wrap-up paragraph providing an overall summary of the key findings. Based on this review, is it possible to identify the most beneficial interventions and for whom? Or does the heterogeneity in measures, methods, populations and outcomes limit any potential conclusions?

15) Section 5 Conclusion - The authors acknowledge in the limitations that there was high heterogeneity across studies, thus, the question remains on what "beneficial" means here. Were there any populations that benefited the most? Is there a geographical bias that may be linked to a cultural bias? Is the recommendation of integrating NBIS in primary care really supported by these findings? I would recommend to reconsider the statement at the end of p.27 in light of these limitations.

Reviewer #2: Manuscript is technically sound. Data presented supports the conclusions, providing summary conclusions for the analyses studies. No specific statistical tests have been conducted, instead summary conclusions have been provided based on summarising study-specific conclusions.

Authors have summarised previously published data and thus data is assumed to be found from the studies used. Summary tables have been provided including the study, variables of interest and outcomes observed.

Manuscript is easy to read and follows a logical order. Figures are in most cases visually appealing, however, in places hard to follow. Whilst the overall message of the figures can be understood from the size of the ‘beneficial effect’ vs ‘non-significant’ component, tracking individual paths is sometimes hard due to size of the paths and crossing of other paths.

Additional comments:

Outlined well the importance/relevance of tackling CVD and cancer i.e., costs to health care etc. and the potential wide ranging benefits of green space and NBI. As most studies analysed were from China and Japan, would be useful to have provided some information (if available), like you did for the US mainly, on the burden of CVD in those countries and whether the higher prevalence of green space and NBI are having any impact on reducing these burdens compared to other countries where such interventions are lacking.

Minor point - could have colour coded the RoB table to make it easier for the reader to gauge RoB (i.e. green, yellow, red)

Section 3.4. first sentence 'king' instead of 'kind'

Section 3.4 Instead of saying 'some studies...' state how many, instead of having to count the number of references provided for the sentence.

Are there any studies measuring or estimating cardiorespiratory fitness as a measure of cardiovascular health following NBI or greenspace activities?

Might be useful for readers to be provided with a definition heart rate variability and the relevance of changes in HRV.

Section 4.1.2 - Identify for reader whether exercising groups without a visual stimuli were used as controls and state whether there was a difference in outcomes between groups. It would be beneficial to have a clear picture of how much benefit on CVD and cancer related health markers there was when a nature visual stimuli is added compared to regular exercise without it.

Conclusions on climate change resilience and climate resilient neighbourhoods was not clear.

6. PLOS authors have the option to publish the peer review history of their article (what does this mean?). If published, this will include your full peer review and any attached files.

Reviewer #1: **Yes: **Marica Cassarino

Reviewer #2: No

---

## [Author Response · Author response to Decision Letter 0]

26 Aug 2022

Reviewer #1: 

The manuscript presents a systematic review of the impact of green spaces and nature-based interventions (NBIs) on cardiovascular and cancer outcomes. The review included 31 experimental studies which indicate a potential beneficial health effect of some form of NBI, although the available evidence presents considerable heterogeneity. The review protocol is registered in PROSPERO and the PRISMA guidelines have been adhered to, with some minor inaccuracies.

While the search appears to have been done rigorously, there are some issues that require addressing, as listed below:

1) Introduction - Sections discussing greenspace would benefit from presenting a definition of this. "Greenspace" refers in the literature to many different things (e.g., urban parks, wild nature, gardens, normalized difference vegetative index) and may at times include blue spaces. What definition, if any, guided this systematic review?

Response: We used an expanded definition of greenspace exposure as described by authors of articles included in our review, including forest bathing, gardening, nature viewing, etc. We clarified this in the manuscript that the definition of greenspace encompasses forest, nature, parks, trees, etc.

2) Introduction - The review includes nature-based interventions as part of the search, but these are not discussed in the Introduction with regards to their definition and rationale for including. What is the added value of looking at NBIs as well as greenspaces? Where do NBIs sit within the MBASIC framework?

Response: The goal of the review was to look at experimental exposure to greenspaces or any other exposure to nature on CVD or cancer related outcomes. That’s why we used the term “nature-based interventions” to maintain that inclusivity. 

3) Introduction - A recent systematic review with meta-analysis looked at the relationship between green spaces and CVD (https://doi.org/10.1016/j.envpol.2022.118990). What does this review add?

Response: Our paper is unique as it specifically looks at experimental studies looking at both CVD or cancer related outcomes to highlight the close links between the two pathologies and . We want to understand what intervention studies have been conducted and help us propose actionable greenspace or nature-based interventions to improve cardiovascular health and cancer outcomes, if any. 

4) Section 2.2 Article selection process- The criteria for intervention should be more clearly described. Any type of green space? Does the "other" means here that you looked at interventions not taking place in green spaces?

Response: The whole idea of this section is to show how we selected the article based on a pre-defined PICO framework. The “intervention” criteria are referring to identifying articles that describe any kind of exposure to greenspace type such as forest bathing, greening exercise, nature viewing, urban parks… which are all nature based interventions. Any experimental exposure to any greenspace is considered in this review. We have removed the word “Other” to help clarify this. 

5) Section 2.2 Article selection process - The criteria for cancer related outcomes described here (lifestyle changes and QoL) do not seem to match the outcomes described in the Introduction as well as section 2.1 (which include cancer prognosis, cancer incidence, cancer mortality, etc.). I would encourage the authors to clarify what cancer-related outcomes were investigated and provide a rationale for looking at these outcomes.

Response: We wanted to look at any cancer related outcomes as described by authors of included studies. This essentially included cancer prognosis, cancer incidence, cancer mortality lifestyle changes, quality of life, etc. We only reported what we found in studies, as reported by authors. Had we found any paper looking at incidence, mortality, etc., we would have reported those findings as well. 

6) Section 2.3 Eligibility Criteria - Table 1 presents as inclusion criterion "empirical studies" but this does not fully reflect the specific designs that appears to have been included in the review, i.e., experimental with or without control and quasi-experimental. Please clarify

Response: Thank you for this comment! We removed the word “empirical” and clarified our inclusion as looking at experimental studies (with or without control) and quasi-experimental

7) Section 2.4 Data extraction - The information presented here does not match the PROSPERO protocol, which is much more comprehensive than the one presented here (see extract from protocol below). I also note that covariates are not discussed in the review, despite this was included in the protocol. Please clarify why data extraction did not adhere to the protocol. 

From the PROSPERO protocol: "We will extract the following data from articles: (1) Studies geographical information (City, state, country); (2) Studies urbanicity setting (rural, semi-urban, or urban); (3) The type of greenspace or nature-based interventions + assumptions made or hypotheses; (4) Measures of any CVD related outcome (Incidence, morbidity, or CVD related mortality); (5) Measures of any cancer related outcome (anything from the cancer control continuum, cancer related quality of life (QOL), or cancer related mortality; (6) CVD or cancer type under investigation (specific or any type); (7) All covariates adjusted for: a. Individual level factors: i. Demographic information (when available); ii. Socioeconomic information (when available); iii. Co-morbidity information (when available). b. Neighborhood factors (when available): i. Social environment factors, ii. Other neighborhood-built, environment characteristics. (8) Statistical analyses conducted; (9) Studies strengths and weaknesses"

Response: Thank you very much for this comment! We edited the relevant tables and the current tables’ content reflect what we proposed in the PROSPERO Protocol. We added names of cities, state and countries if clarified in included papers, along with urbanicity setting for each study. We also added the column on hypothesis or assumptions made by authors of included papers, and control group, and provided information. We finally added the wording on “greenspace type”. We also added a column on covariate and discussed all changes in the manuscript. Every change made is highlighted with track changes. By making these changes, the review is adhering to the published protocol in PROSPERO. 

8) PRISMA flowchart: In the screening phase, please clarify the criteria for excluding 45 abstracts.

Response: 45 abstracts were excluded because they did not meet at least one of our pre-defined inclusion criteria. Each of the excluded studies was either not experimental, or not looking at one of the outcomes of interest. 

9) Risk of bias: The authors appear to have used the NOS for cohort studies, but given the inclusion of experimental studies only, it is unclear why a more appropriate tool, like the ROB2 or ROBINS-I, was not used. The authors discuss this in the limitations, but the reason for not using other tools beyond NOS should be clarified.

Importantly, the criterion used for assessing the representativeness of the exposed group does not match the criterion set in the NOS scale. Here, representativeness should relate to the community where the study took place. Judging representativeness based on whether the sample used in the study matches the title/abstract can be prone to bias per se. I would strongly encourage the authors to reassess this criterion and provide a clearer justification for the choice of tool to assess risk of bias.

Response: The ROB2 is specifically used for examining the risk of bias in randomized trial studies, while ROBINS-I is for non-randomized studies of exposures. We chose the NOS tool as the proposed alternative as it has been validated for case-control and cohort studies because our experimental arms can be treated as cases, while controls groups be treated as “control”. Studies that use repeated measures (i.e., single arm without a control group, with a before and after exposure measurements) were treated as cohort, and therefore this processes eased the use of the NOS tool. 

We reassessed the representativeness criterion by specifying out intentional to ensure that a score was given if the group truly represents the described group in the title or abstract (or somewhat representative of it). We made that change in the criterion description. 

10) Section 3.4 Study design and demographics - Were the samples included in the studies composed of healthy individuals or patients populations? While this may be self-evident for the cancer studies, it is unclear for CVD and it would be useful to tease out whether the interventions work as a preventative measure (i.e., maintaining good CVH among healthy individuals) or remedy (improving outcomes among people with CVD).

Response: Table 3 and 4 reports sample size considered for included studies. Since the overall goal of the review is to look at the impact of interventions on outcomes, both preventive (outcomes related at ensuring good CVH) and those intended to reduce the burden of CVD among individuals with CVD are all considered. That’s why we coded the alluvial charts with either “Beneficial effect” or “Not significant” or “Significant in controls”. This help us illustrate the trends in findings when all included studies are taken together. This has important clinical and public health implications for the use of noninvasive greenspace and nature-based interventions in improving CVD and/or cancer outcomes. 

11) Sections 3.7.1 and 3.7.2 - Two main comments here: 1) The types of activities completed by the control groups (if present) should be discussed, as it is unclear what the NBIs were compared to; this should also be evident in Table 3 and Table 4, which should indicate whether a control group was included and what they did. 2) It is unclear how the reviewers decided on a beneficial vs. nonsignificant effect. This requires better clarification. Were effect sizes considered for this? Also, did any of the studies find a negative effect of green space or NBIs?

Response: In Tables 3 and 4, we added specific information on control groups, whether it was a pre-post study design without a control group, or what the group groups did when present. The presence of control groups or not is also one of the components considered for the risk of bias assessment. The conclusion on whether the intervention was beneficial or not was based on studies findings and conclusion as compared to a desirable outcomes (reduction on blood pressure if considered beneficial among hypertensive patients, or gardening continuation is considered beneficial as it is a good human behavior linked with positive health outcomes. Since we did not do a meta-analyses, no effect sizes were measured, nor reported. We only reported trends observed in alluvial charts. 

12) Figure 2: I acknowledge the effort made by the authors in this visual depiction, but I must admit that it requires quite a lot of effort to make sense of. For instance, it is unclear what criterion determined the thickness of each study. Should the studies be organized based on region of the world or continent rather than country? Could a pattern or colour code like in Figure 3 be used to distinguish beneficial effect from "not significant"?

Response: Alluvial charts intention is to show findings trends across multiple criteria being considered all together. The question about the “thickness for each study” is indicative of how many statistical tests were conducted to test a particular hypothesis relevant to at least one of the outcomes considered for this study. Each test represents a single line as it can be seen in the Appendices C1 and C2, attached with this submission. Alluvial charts allowed us to visually comprehensively present all statistical tests conduced by all studies and what the trends in findings are by significant “beneficial effects” or “Not significant” or “Significant in control, which indicates that changes were seen in controls, but not in experimental groups, which does not in any way indicates that greenspace or nature based interventions have adverse outcomes in those context. That’s why we chose to report what the study found. 

On the question about organizing studies based on the region, it is possible, but not the intention of the chart we wanted to produce. The color pattern in figure 3 helps to easily show trends since it has less quantities to be reported, something that would rather be confusing in figure 2 that has lots of quantities to showcase. 

13) Section 3.7.2 and Figure 3 - What is defined here as "significant in control only" deserves better clarification. Does this imply inferiority of the NBI compared to the control, i.e., a negative effect of the intervention? Or is this related to no change observed in the intervention?

Response: “Significant in control only” means that expected benefits of the interventions were seen in controls instead of being seen in the interventional groups. It does not in any way suggest that any negative effect of the intervention. 

14) Discussion - This section would benefit from a wrap-up paragraph providing an overall summary of the key findings. Based on this review, is it possible to identify the most beneficial interventions and for whom? Or does the heterogeneity in measures, methods, populations, and outcomes limit any potential conclusions?

Response: We added a wrap up paragraph as suggested. In the paragraph, we referred to intervention-specific alluvial charts to suggest interventions that had most beneficial outcomes but highlighted the patient-centric approach to ensure individual preference as we thrive to improve CVD and/or cancer related outcomes. 

15) Section 5 Conclusion - The authors acknowledge in the limitations that there was high heterogeneity across studies, thus, the question remains on what "beneficial" means here. Were there any populations that benefited the most? Is there a geographical bias that may be linked to a cultural bias? Is the recommendation of integrating NBIS in primary care really supported by these findings? I would recommend reconsidering the statement at the end of p.27 in light of these limitations.

Response: In the limitations section, we have highlighted that presented trend across all studies should be seen as a descriptive summary of findings; and any inference made should consider all studies collectively because no meta-analysis was done. It is still an accurately description of all studies considered collectively as seen in the alluvial charts. 

Reviewer #2: 

Manuscript is technically sound. Data presented supports the conclusions, providing summary conclusions for the analyses studies. No specific statistical tests have been conducted, instead summary conclusions have been provided based on summarising study-specific conclusions.

Authors have summarised previously published data and thus data is assumed to be found from the studies used. Summary tables have been provided including the study, variables of interest and outcomes observed.

Manuscript is easy to read and follows a logical order. Figures are in most cases visually appealing, however, in places hard to follow. Whilst the overall message of the figures can be understood from the size of the ‘beneficial effect’ vs ‘non-significant’ component, tracking individual paths is sometimes hard due to size of the paths and crossing of other paths.

Additional comments:

Outlined well the importance/relevance of tackling CVD and cancer i.e., costs to health care etc. and the potential wide ranging benefits of green space and NBI. As most studies analysed were from China and Japan, would be useful to have provided some information (if available), like you did for the US mainly, on the burden of CVD in those countries and whether the higher prevalence of green space and NBI are having any impact on reducing these burdens compared to other countries where such interventions are lacking.

Response: Thank you for the comment. Actually, most studies included in our review were conducted Asia for the great part (China and Japan). There is a dearth of studies on greenspace or NBI and CVD or cancer in other countries, which could offer comparative information on the financial impact and effects in reducing the burden of those diseases in other under studied contexts. 

Minor point - could have colour coded the RoB table to make it easier for the reader to gauge RoB (i.e. green, yellow, red)

Response: We thought it was best to use the annotation used by other previous scholars by assigning score or stars and then count the total to provide an overall score for the risk of bias. 

Section 3.4. first sentence 'king' instead of 'kind'

Response: We have corrected the wording “king” and changed into “kind”

Section 3.4 Instead of saying 'some studies...' state how many, instead of having to count the number of references provided for the sentence.

Response: We have addressed this comment and reported the exact number of studies for each point being reported on. 

Are there any studies measuring or estimating cardiorespiratory fitness as a measure of cardiovascular health following NBI or greenspace activities?

Response: In the review did not find any study reporting on cardiorespiratory fitness as a measure of cardiovascular health following NBI or greenspace activities, but if we did, they would have been considered as they would have met our pre-defined inclusion criteria. 

Might be useful for readers to be provided with a definition heart rate variability and the relevance of changes in HRV.

Response: When we first introduced the concept of HRV in Table 3, we defined it as the intervals between successive heartbeats.

Section 4.1.2 - Identify for reader whether exercising groups without a visual stimuli were used as controls and state whether there was a difference in outcomes between groups. It would be beneficial to have a clear picture of how much benefit on CVD and cancer related health markers there was when a nature visual stimuli is added compared to regular exercise without it.

Response: We highlighted that green exercise was found to be positively associated with many outcome measures related to CV health with few statistical tests that found no significant associations or no numerical difference at all. This is true for visual stimuli studies. Many tests showed beneficial effects while some tests showed no difference. 

Conclusions on climate change resilience and climate resilient neighbourhoods was not clear.

Response: When we highlighted the role of greenspace or NBI on CVD and/or cancer related outcomes, we also mentioned their critical contribution to climate resilient neighborhoods.

---

## [Decision Letter · Decision Letter 1]

22 Sep 2022

PONE-D-22-07911R1The Impact of greenspace or nature-based interventions on cardiovascular health or cancer related outcomes: A systematic review of experimental studiesPLOS ONE

Dear Dr. BIKOMEYE,

Thank you for submitting your manuscript to PLOS ONE. After careful consideration, we feel that it has merit but does not fully meet PLOS ONE’s publication criteria as it currently stands. Therefore, we invite you to submit a revised version of the manuscript that addresses the points raised during the review process.

Please include some justifications presented in the response to reviewer 1 in to the manuscript to allow us to accept the manuscript. Reviewer 2 comments on Figure 2, but this isn't a necessary amendment - however if you are able to make the figure slightly clearer feel free to do so. 

We look forward to receiving your revised manuscript.

Kind regards,

Lindsay Bottoms

Academic Editor

PLOS ONE

Journal Requirements:

Additional Editor Comments:

Dear Authors,

Thanks for going through the previous suggestions and responding to the concerns of the reviewers, in particular reviewer 1. However, reviewer 1 points out that some changes still need to be made to the manuscript to justify the points you make well in the response. I agree with the reviewer and would kindly ask you make some minor amendments to the manuscript to accound for these justifications.

Thanks,

Reviewers' comments:

Reviewer's Responses to Questions

**Comments to the Author**

1. If the authors have adequately addressed your comments raised in a previous round of review and you feel that this manuscript is now acceptable for publication, you may indicate that here to bypass the “Comments to the Author” section, enter your conflict of interest statement in the “Confidential to Editor” section, and submit your "Accept" recommendation.

Reviewer #1: (No Response)

Reviewer #2: All comments have been addressed

2. Is the manuscript technically sound, and do the data support the conclusions?

Reviewer #1: Yes

Reviewer #2: Yes

3. Has the statistical analysis been performed appropriately and rigorously? 

Reviewer #1: Yes

Reviewer #2: Yes

4. Have the authors made all data underlying the findings in their manuscript fully available?

Reviewer #1: Yes

Reviewer #2: Yes

5. Is the manuscript presented in an intelligible fashion and written in standard English?

Reviewer #1: Yes

Reviewer #2: Yes

6. Review Comments to the Author

Reviewer #1: Most comments have been addressed. I note that the authors provided an appropriate reply to comments #3, 4, 5, 8, 10, and 13. The reason for providing those comments is that that information should be made more evident in the paper. However, it would appear that the authors have addressed the answers without making any changes to the manuscript. This may cause a reader to have the same doubts as the present reviewer.

For example, in comment 3 the authors were encouraged to discuss the novelty of this review compared to a previous similar systematic review. (https://doi.org/10.1016/j.envpol.2022.118990). The authors answer is satisfactory in that it explains clearly the point of novelty. However, this is nowhere to be found in the paper. Thus, a potential reader who has read the other systematic review before may be left in doubt as to what this review adds compared to the previous. Hence, my recommendation to discuss this point explicitly in the paper.

The same applies for the other comments. The authors' answers are clear, but they do not seem to be reflected in the manuscript revisions.

Reviewer #2: Regarding comments put forward by reviewer #2, these have all been addressed and amendments and recommendations have been considered when revising the manuscript. #

Some minor comments would be that some of the figures I personally feel are too hard to follow (e.g., Figure 2), I appreciate what the figure shows and some figures are a lot easier to follow however with the amount of information in figure 2 I wonder if another visual representation could be used. The brief description of what the line thickness represents needs to come much earlier in the paper not just in the last paragraph of the discussion.

Finally, section 4.1.1 in the discussion is very descriptive for a 'discussion' section. As it stands this would sit better in a results section, adding more discussion elements here would greatly benefit the opening section of the discussion.

7. PLOS authors have the option to publish the peer review history of their article (what does this mean?). If published, this will include your full peer review and any attached files.

Reviewer #1: **Yes: **Marica Cassarino

Reviewer #2: No

---

## [Author Response · Author response to Decision Letter 1]

6 Oct 2022

Dr. Bottoms: 

Thank you for pointing out additional concerns for reviewer 1. Reviewer 1 suggest that we ensure that every explanation given in comments is explicitly addressed in the manuscript. We are hereby adding more comments # 3, 4, 5, 8, 10 and 15 as pointed out. Those clarifications are indicated in italics. 

Comment # 3) Introduction - A recent systematic review with meta-analysis looked at the relationship between green spaces and CVD (https://doi.org/10.1016/j.envpol.2022.118990). What does this review add?

Response: As note in our initial response, our paper is unique as it specifically looks at experimental studies looking at both CVD and/or cancer related outcomes to highlight the close links between the two pathologies. We want to understand what intervention studies have been conducted and help us propose actionable greenspace or nature-based interventions to improve cardiovascular health and cancer outcomes, if any. 

We have expanded last paragraph of our introduction to include information about the focus on our review that looks at “experimental studies only” and have also cited most recent review on greenspace and CVD outcomes. In previous paragraph of our manuscript, we had highlighted the close links between the two pathologies. 

Comment # 4) Section 2.2 Article selection process- The criteria for intervention should be more clearly described. Any type of green space? Does the "other" means here that you looked at interventions not taking place in green spaces?

Response: The whole idea of this section is to show how we selected the article based on a pre-defined PICO framework. The “intervention” criteria are referring to identifying articles that describe any kind of exposure to greenspace type such as forest bathing, greening exercise, nature viewing, urban parks… which are all nature based interventions. Any experimental exposure to any greenspace is considered in this review. We have removed the word “Other” to help clarify this. 

We added examples of potential greenspace interventions indicating the type of exposure to greenspace, including forest bathing, greening exercise, nature viewing, and gardening. 

5) Section 2.2 Article selection process - The criteria for cancer related outcomes described here (lifestyle changes and QoL) do not seem to match the outcomes described in the Introduction as well as section 2.1 (which include cancer prognosis, cancer incidence, cancer mortality, etc.). I would encourage the authors to clarify what cancer-related outcomes were investigated and provide a rationale for looking at these outcomes.

Response: We wanted to look at any cancer related outcomes as described by authors of included studies. This essentially included cancer prognosis, cancer incidence, cancer mortality, lifestyle changes during cancer survivorship, quality of life, etc. We only reported what we found in studies, as reported by authors. Had we found any paper looking at incidence, mortality, etc., we would have reported those findings as well. 

We offered an expanded explanation on cancer related outcomes including those named in our previous response. 

8) PRISMA flowchart: In the screening phase, please clarify the criteria for excluding 45 abstracts.

Response: 45 abstracts were excluded because they did not meet at least one of our pre-defined inclusion criteria. Each of the excluded studies was either not experimental, or not looking at one of the outcomes of interest. 

We added the explanation indicating the reason of exclusion of 45 abstracts in the text. This new sentence has been added: “At the abstract screening stage, 45 articles were excluded because they did not meet at least one of our pre-defined inclusion criteria. Each one of the excluded studies was either not experimental, or not looking at one of the outcomes of interest.” We also made those changes in the PRISMA Chart 

10) Section 3.4 Study design and demographics - Were the samples included in the studies composed of healthy individuals or patients populations? While this may be self-evident for the cancer studies, it is unclear for CVD and it would be useful to tease out whether the interventions work as a preventative measure (i.e., maintaining good CVH among healthy individuals) or remedy (improving outcomes among people with CVD).

Response: Table 3 and 4 reports sample size considered for included studies. Since the overall goal of the review is to look at the impact of interventions on outcomes, both preventive (outcomes related at ensuring good CVH) and those intended to reduce the burden of CVD among individuals with CVD are all considered. That’s why we coded the alluvial charts with either “Beneficial effect” or “Not significant” or “Significant in controls”. This help us illustrate the trends in findings when all included studies are taken together. This has important clinical and public health implications for the use of noninvasive greenspace and nature-based interventions in improving CVD and/or cancer outcomes.

We explained the composition of population sample in the text. It was explained in section 2.2. Article selection process. There was no restriction to the population in terms of demographic or disease status. We also add more clarification CVD outcomes measures. This sentence has been added in 2.2. as well: “Both preventive measures (indictors of good CV health among healthy individuals) and restorative measures (indicators of improved CV health among individuals with CVD) are all considered.”

---

## [Editor Report · Decision Letter 2]

10 Oct 2022

The Impact of greenspace or nature-based interventions on cardiovascular health or cancer related outcomes: A systematic review of experimental studies

PONE-D-22-07911R2

Dear Dr. BIKOMEYE,

We’re pleased to inform you that your manuscript has been judged scientifically suitable for publication and will be formally accepted for publication once it meets all outstanding technical requirements.

Kind regards,

Lindsay Bottoms

Academic Editor

PLOS ONE

Additional Editor Comments (optional):

Thank you for adding the justifications to the manuscript and making the minor amendments suggested by the reviewers. I am happy to accept the manuscript in its current form.
---

## [Editor Report · Acceptance letter]

27 Oct 2022

PONE-D-22-07911R2 

The Impact of greenspace or nature-based interventions on cardiovascular health or cancer-related outcomes: A systematic review of experimental studies 

Dear Dr. Beyer:

I'm pleased to inform you that your manuscript has been deemed suitable for publication in PLOS ONE. Congratulations! Your manuscript is now with our production department. 

Kind regards, 

on behalf of

Dr. Lindsay Bottoms 

Academic Editor

PLOS ONE